# FRUSTRATINGLY SIMPLE RETRIEVAL IMPROVES CHALLENGING, REASONING-INTENSIVE BENCHMARKS

**Xinxi Lyu**[*i]   **Michael Duan**[*s]   **Rulin Shao**[wa]   **Pang Wei Koh**[wa]   **Sewon Min**[ba]

[i]University of Illinois Urbana-Champaign   [s]University of Southern California
[a]Allen Institute for AI   [w]University of Washington   [b]University of California, Berkeley

seanlyu2@illinois.edu   duanm@usc.edu   sewonm@berkeley.edu

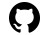 alrope/CompactDS-102GB   ⬛ compactds-retrieval

## ABSTRACT

Retrieval-augmented Generation (RAG) has primarily been studied in limited settings, such as factoid question answering; more challenging, reasoning-intensive benchmarks have seen limited success from minimal RAG. In this work, we challenge this prevailing view on established, reasoning-intensive benchmarks: MMLU, MMLU Pro, AGI Eval, GPQA, and MATH. We identify a key missing component in prior work: a usable, web-scale datastore aligned with the breadth of pretraining data. To this end, we introduce COMPACTDS: a diverse, high-quality, web-scale datastore that achieves high retrieval accuracy and subsecond latency on a single-node. The key insights are (1) most web content can be filtered out without sacrificing coverage, and a compact, high-quality subset is sufficient; and (2) combining in-memory approximate nearest neighbor (ANN) retrieval and on-disk exact search balances speed and recall. Using COMPACTDS, we show that a minimal RAG pipeline achieves consistent accuracy improvements across all benchmarks and model sizes (8B–70B), with relative gains of 10% on MMLU, 33% on MMLU Pro, 14% on GPQA, and 19% on MATH. No single data source suffices alone, highlighting the importance of diversity of sources (web crawls, curated math, academic papers, educational text). Finally, we show that our carefully designed in-house datastore matches or outperforms web search engines such as Google Search, as well as recently proposed, complex agent-based RAG systems—all while maintaining simplicity, reproducibility, and self-containment. We release COMPACTDS and our retrieval pipeline, supporting future research exploring retrieval-based AI systems.

## 1 INTRODUCTION

Retrieval-based language models (LMs) enhance generation by retrieving relevant information from a large text datastore and feeding it into the model (Guu et al., 2020; Karpukhin et al., 2020; Lewis et al., 2020; Borgeaud et al., 2022; Shi et al., 2023; Asai et al., 2023; Shao et al., 2024). This approach has proven highly effective for tasks like question answering and fact verification, especially for information-seeking queries that require precise facts that can be found in a highly curated knowledge source like Wikipedia (Kwiatkowski et al., 2019; Lee et al., 2019; Joshi et al., 2017; Petroni et al., 2020). However, much less is understood about the utility of retrieval beyond factoid tasks. Prior work suggests that retrieval offers no benefit and even hurts performance on reasoning-intensive tasks (BehnamGhader et al., 2022; Geng et al., 2024). Some studies have introduced agentic pipelines to address this gap, but they either rely on web search engines (Li et al., 2025b; Wu et al., 2025) or are limited to Wikipedia (Jin et al., 2025; Chen et al., 2025; Song et al., 2025; Sun et al., 2025).

In this work, we revisit this assumption and ask: How far can we improve performance on challenging, reasoning-intensive LM benchmarks using *minimal retrieval—dense retrieval followed by generation*?

---

*Equal contribution.

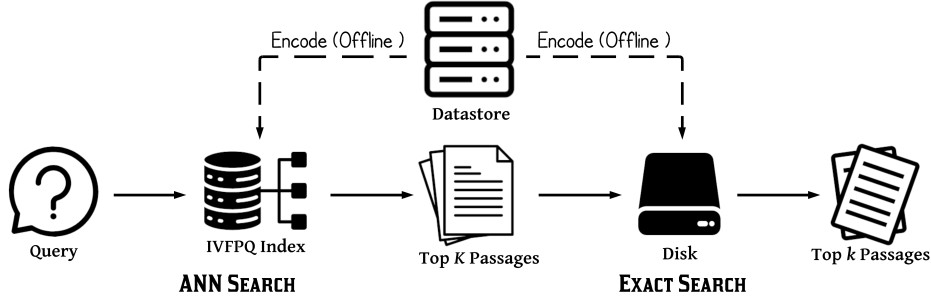

Figure 1: An illustration of the two-stage dense retrieval pipeline with COMPACTDS.

To this end, we evaluate a minimal retrieval setup across a diverse set of established benchmarks, including MMLU (Hendrycks et al., 2021a), MMLU Pro (Wang et al., 2024), AGI Eval (Zhong et al., 2023), GPQA (Rein et al., 2023), and MATH (Hendrycks et al., 2021b). We identify a critical limitation in prior work: the absence of an accessible general-purpose datastore. Prior work used either Wikipedia-based datastores, which are limited in coverage on these general-purpose benchmarks (as we show in §4), or large-scale web datastores that contain massive unfiltered low-quality sources and are inaccessible in practice (Borgeaud et al., 2022; Shao et al., 2024), e.g., requiring over 12TB of RAM unless deployed on a distributed serving system.

To this end, we introduce **COMPACTDS**, a 380-billion-word datastore constructed from a high-quality, diverse collection of sources—including web crawls, curated math content, academic papers, and educational text—designed to match the breadth of pretraining data while remaining accessible. COMPACTDS is built on two key insights. First, it is possible to aggressively filter low-quality web text while preserving the coverage and diversity of the web, resulting in a smaller yet representative dataset. Second, combining in-memory approximate nearest neighbor (ANN) search (Johnson et al., 2019) with on-disk exact inner product search enables subsecond retrieval on a single 456GB RAM node—far more efficient than prior systems that require several TBs of RAM.

COMPACTDS leads to significant performance improvements on benchmarks that were previously thought not to benefit from minimal retrieval: +10% on MMLU, +33% on MMLU Pro, +14% on GPQA, and +19% on MATH with Llama 3.1 8B Instruct, with persistent gains as the model scales to 70B parameters. Notably, exact inner product search with a more expressive encoder following ANN helps compared to using ANN alone, widening gains over no retrieval from 26% to 33% on MMLU Pro and from 14% to 19% on MATH. Our ablation study on datastore composition reveals several key insights: (1) Dataset diversity is critical: no single data source suffices alone, and even removing weakest contributing sources leads to degraded performance; (2) Including educational materials, often absent from web crawls, consistently improves performance; and (3) Quality filtering methods developed for pretraining generally help.

Notably, our in-house datastore matches or outperforms web search engines like Google Search across all benchmarks—a result that was previously unattainable using standard RAG benchmarks built with web search as an oracle (Kwiatkowski et al., 2019). It also matches or outperforms recently proposed, complex systems based on web search and agentic pipelines (Li et al., 2025b), as demonstrated on GPQA Diamond (Rein et al., 2023) and MATH-500 (Hendrycks et al., 2021b) using QwQ 32B.

We release COMPACTDS and our retrieval pipeline as a reproducible, academically accessible alternative to web search. We hope COMPACTDS facilitates future work on retrieval-based AI systems, including integration with agentic pipelines and training with retrieval.

## 2 RELATED WORK

**Retrieval Augmentation and Evaluation.**  Retrieval augmented generation (RAG) has shown remarkable success in factoid question-answering tasks (Karpukhin et al., 2020; Lewis et al., 2020). However, the majority of RAG benchmarks focus on fact retrieval, typically supported by highly curated knowledge sources such as Wikipedia (Kwiatkowski et al., 2019; Lee et al., 2019; Joshi et al., 2017; Petroni et al., 2020); the applicability of RAG to tasks beyond this scope remains largely unexplored. In fact, prior work presented that RAG is ineffective for other tasks like reasoning tasks (BehnamGhader et al., 2022; Geng et al., 2024) and open-ended generation (Wang et al., 2023).

In this work, we evaluate retrieval on a broad set of benchmarks—MMLU, MMLU Pro, AGI Eval, GPQA, and MATH—that are well-established and not specifically designed for RAG, and known to be reasoning-intensive. Concurrent work, ReasonIR (Shao et al., 2025), also demonstrates the value of retrieval for reasoning tasks, but takes an orthogonal approach: ReasonIR focuses on improving the embedding model, whereas we focus on improving the datastore and nearest neighbor search.

**Improving Datastores.** While extensive work has focused on improving *training* datasets (Raffel et al., 2020; Soldaini et al., 2024; Li et al., 2024; Penedo et al., 2024), little attention has been given to improving data as a retrieval datastore. Beyond Wikipedia, some efforts have used broader web sources, e.g., a small random subset of Common Crawl (five billion tokens) (Shi et al., 2023; Lin et al., 2023; Piktus et al., 2021). These efforts were still evaluated on perplexity or Wikipedia-based benchmarks (except Shi et al. (2023) on MMLU, which we compare against).

We argue that prior datastores are either too narrow or small to be broadly effective, or not practically usable, e.g., MassiveDS (Shao et al., 2024) requires over 12TB of RAM to avoid multi-minute latency, making deployment infeasible in typical academic settings without distributed infrastructure. This work directly addresses these issues, proposing a datastore that is large and broad in coverage, yet compact enough to enable subsecond latency in a single-node deployment.

**Agentic RAG.** Recently, agentic RAG, which iteratively issues search queries, retrieves information, and reasons over results to perform reasoning-intensive tasks, has emerged as an active area of research. These approaches can be broadly divided into two categories: (1) prompt-based methods that do not require training (Li et al., 2025b; Wu et al., 2025), and (2) training-based methods that fine-tune a reasoning LM to use search, typically via reinforcement learning (Jin et al., 2025; Chen et al., 2025; Song et al., 2025; Sun et al., 2025). Much of this work uses web search engines, which are costly, hard to reproduce, and unstable, making them unsuitable for training, as also noted by (Sun et al., 2025). Consequently, most training-based work uses an in-house Wikipedia datastore and only evaluate on Wikipedia-based benchmarks.

Instead of optimizing for agentic RAG, our work focuses on minimal RAG, which is a fundamental building block of any retrieval-based AI systems that can be easily integrated. This agentic RAG literature, however, highlights an emerging need for high-quality, general-purpose in-house datastores, particularly to enhance reproducibility, improve stability, and ensure cost efficiency.

## 3 METHOD

We discuss two key ideas that enable a high-quality, high-coverage retrieval datastore: data sources that match the breadth of pretraining corpora while filtering out low-quality web text (§3.1), and approximate nearest neighbor (ANN) search followed by exact search (§3.2).

### 3.1 COMPACTDS DATA SOURCES

To match the breadth of pretraining corpora while achieving high quality and diversity, we strategically construct COMPACTDS with the following data sources:

**Web Crawl.** To ensure wide coverage, we start with Common Crawl, which is widely used for pre-training and also constitutes 70% of MASSIVEDS (Shao et al., 2024). However, we hypothesize that much of it is low-quality and unnecessary for retrieval. Therefore, we construct a compact, high-quality subset (**High-quality CC**) using a series of filtering steps. We take the union of C4 (Raffel et al., 2019), a small curated subset, and DCLM-Baseline (Li et al., 2025a), which has undergone extensive manual and model-based filtering, and then further filter DCLM-baseline using the FineWeb-Edu classifier (Penedo et al., 2024) with a threshold of 4.0, which filters text based on its educational value. This process reduces the size of Common Crawl from 894B words to 172B words.

**Wikipedia and Books.** We then augment the web crawl with data sources commonly regarded as high quality for pretraining: Wikipedia, books, and educational text.

- ***Wikipedia***: We consider two sources: Wikipedia from DPR (Karpukhin et al., 2020) and 2) Wikipedia in RedPajama-V1 (Computer, 2023).
- ***Books***: We use the RedPajama-V1 Books subset, containing a wide variety of digitized eBooks.
- ***Educational Text***: We take the data from Shi et al. (2025), text extracted from digitized PDFs.

**Expert Data.** To broaden the coverage, we include more data sources that contain expertise knowledge in areas such as math, science, and coding:

- *Math*: We combine OpenWebMath (Paster et al., 2023), a filtered math webpages from Common Crawl, and NaturalProofs (Welleck et al., 2021), a corpus including theorems, proofs, definitions, and related content.
- *Academic Papers*: We consider three sources: Pes2o (Soldaini & Lo, 2023), open-access academic papers, PubMed (National Library of Medicine, 2023), a collection of biomedical and life sciences journal literature, and ArXiv (Computer, 2023).
- *Github*: We use the RedPajama-V1 GitHub subset, a well-established source for code.

**Q&A Forums.** Finally, we include Q&A forums widely used in pretraining:

- *Stack Exchange*: We use the RedPajama-V1 subset of Stack Exchange, a collection of high-quality, community-sourced Q&A spanning diverse domains from computer science to chemistry.
- *Reddit*: We take the Reddit data from (Shi et al., 2025).

**Decontamination.** To ensure the robustness of our evaluation, we filter out paragraphs with an over 70% 13-gram Jaccard similarity with any query in all of our evaluation datasets following Borgeaud et al. (2022). The impact of this decontamination method on performance is reported in §A.1.

In total, COMPACTDS includes 380.5 billion words from 639 million documents. Each document is split into passages of 256 words, following (Shao et al., 2024), resulting in 1.9 billion passages. More details and statistics are provided in §A.2.

## 3.2 DENSE RETRIEVAL WITH COMPACTDS

COMPACTDS is based on a dense retrieval model over $N$ passages, $\{p_1, p_2, \cdots, p_N\}$. Given an encoder $E$ that maps text into an $h$-dimensional vector, passage embeddings $\mathbf{p}_1, \cdots, \mathbf{p}_N \in \mathbb{R}^h$ are pre-computed and indexed using a nearest neighbor search structure. At test time, a test query $q$ is encoded as $\mathbf{q} = E(q) \in \mathbb{R}^h$, and the top-$k$ passages are retrieved according to $\arg\mathrm{Top}k_{1 \le i \le N} \mathbf{q}^\mathsf{T}\mathbf{p}_i$.

The key challenge, when it comes to deploying web-scale datastores, is the memory and compute overhead in building and serving a nearest neighbor search index for billions of high-dimensional vectors. For example, with $N = 1.9 \times 10^9$ and $h = 768$, storing the passage embeddings in full precision requires $1.9 \times 10^9 \times 768 \times 4 = 5.4\mathrm{TB}$ of vector data, which is infeasible to store in memory.

**Approximate Nearest Neighbor (ANN) Search via IVFPQ.** To reduce search latency and storage requirement, we use approximate nearest neighbor (ANN) via Inverted File with Product Quantization (IVFPQ; Jegou et al. (2010)). IVFPQ partitions the vector space into clusters and quantizes vectors within each cluster to reduce memory usage and improve speed. This allows us to build and serve the ANN index within 456GB of RAM, achieving subsecond retrieval latency. However, IVFPQ typically incurs notable performance degradation due to its lossy nature.

**Second-stage Exact Inner Product Search.** To recover performance lost from the approximate search, we leverage a second-stage exact inner product search. Specifically, the ANN index retrieves a candidate set of $K$ passages ($K \gg k$), which are then re-ranked using their original (non-quantized) embeddings to identify the final top-$k$ results. These exact embeddings can be stored on disk, enabling efficient disk-based search under reasonable disk I/O constraints, especially when $K$ is modest (e.g., $100 \le K \le 1000$). If disk-based search is infeasible, embeddings can alternatively be recomputed on the fly, trading off latency. Moreover, the encoder used for exact search can differ from the one used for ANN, enabling the use of a more expressive model that may be difficult to index. This design choice is inspired by DiskANN (Jayaram Subramanya et al., 2019), which similarly performs in-memory ANN search followed by on-disk exact search.

**Summary.** Altogether, dense retrieval in COMPACTDS proceeds as follows (also shown in Figure 1). Offline, each passage $p_i$ is encoded using $E_{\mathrm{Approx}}$ and indexed with IVFPQ for fast search in RAM (456GB). In parallel, passages are encoded using a stronger encoder, $E_{\mathrm{Exact}}$, and the resulting embeddings are stored on disk. At test time, a query $q$ is encoded as $E_{\mathrm{Approx}}(q)$ and used to retrieve $K$ passages from the IVFPQ index. Then, $q$ is encoded as $E_{\mathrm{Exact}}(q)$, and exact inner product search is applied over the $K$ disk-stored embeddings to obtain the final top-$k$ passages: $\hat{p}_1, \cdots, \hat{p}_k$. We use

CONTRIEVER-MSMARCO (Izacard et al., 2021) as $E_{\text{Approx}}$, and GRITLM-7B (Muennighoff et al., 2024) as $E_{\text{Exact}}$.

This two-stage dense retrieval resolves the memory and latency issue that single-stage exact nearest neighbor search encounters when deploying web-scale datastores, with performance largely restored.

## 3.3  AUGMENTATION WITH COMPACTDS

Given $\hat{p}_1, \hat{p}_2, \cdots \hat{p}_k$ returned by dense retrieval, we feed them into an LLM—denoted as a generator—to generate the answer. We adopt a straightforward augmentation strategy by concatenating the top-$k$ retrieved passages in reverse order, placing the most relevant passage closest to the query $q$, followed by $q$ itself, and feeding it to the LM that generates the answer. We also consider an optional, LLM-based reranking; details are provided in §A.3.

**Oracle Reranking.**  Despite the strong capabilities of LLMs as generators and reranks, they may still struggle to process and reason over a set of retrieved passages, particularly if not explicitly post-trained to do so (Jin et al., 2024). We aim to quantify the upperbound performance achievable from COMPACTDS outputs by defining an *oracle* reranking method. Given a test query $q$ and its ground truth answer $a$, we select the top $k_{\text{rerank}}$ passages from the $K$ candidates retrieved from COMPACTDS-ANN, where $K \geq \hat{k}_{\text{rerank}} \gg k$. Each passage is scored by a generator LLM, based on the increase in the model's likelihood of the gold answer $a$ when appended to the query. The top-$\hat{k}_{\text{rerank}}$ passages based on the oracle scores are put into the generate stage (see details in §C.6).

## 4  EXPERIMENT

### 4.1  EVALUATION SETUP

We evaluate COMPACTDS using the following five benchmarks (detailed in §B): **MMLU** (Hendrycks et al., 2021a) and **MMLU Pro** (Wang et al., 2024), evaluating general knowledge and reasoning over a broad spectrum of subjects (MMLU is grouped into four main categories: STEM, Humanities, Social Sciences, and Other); **AGI Eval** (Zhong et al., 2023), consisting of questions form human-centric standardized exams (e.g., LSAT); **GPQA** (Rein et al., 2023), comprising expert-written, graduate-level multiple-choice questions in biology, physics, and chemistry, and specifically designed to be web-search-proof; and **MATH** (Hendrycks et al., 2021b), a collection of challenging, competition mathematics problems, sourcing problems from exams such as AMC 12 and AIME. The MMLU, MMLU Pro, and AGI Eval datasets are evaluated as multiple choice, while we use chain-of-thought for GPQA and MATH. These are well-established, standard benchmarks in the field, frequently reported by frontier models (OpenAI et al., 2024; Dubey et al., 2024). They are known for their domain diversity and reasoning demands, posing greater challenge compared to the simpler factoid-based QA tasks commonly used in RAG literature (Kwiatkowski et al., 2019; Joshi et al., 2017).

In addition, to compare with Search-o1, a recently proposed agentic RAG system (Li et al., 2025b), we include two benchmarks used in their work (detailed in §5.2): **GPQA Diamond** (Rein et al., 2023), a high-quality subset of GPQA, and **MATH-500** (Lightman et al., 2023), a subset of MATH.

**Language Models.**  We use Llama 3.1 8B Instruct (Dubey et al., 2024) as the default generator for the main experiments and most of the ablations. To evaluate COMPACTDS across different model sizes and families with stronger capabilities, we also include Llama 3.3 70B Instruct, Mistral 7B Instruct (Jiang et al., 2023), Qwen3 8B (Yang et al., 2025), and QwQ 32B (Team, 2024).

### 4.2  MAIN RESULTS

Table 1 presents our main results using Llama 3.1 8B Instruct. COMPACTDS significantly improves performance across all datasets, e.g., 11.0% on MMLU STEM, 33.4% on MMLU Pro, 19.2% on MATH, and 36.2% on GPQA Physics. These improvements are particularly notable given the reasoning-intensive nature of these tasks, e.g., GPQA is specifically designed to be web-search-proof.

**Diversity in COMPACTDS Composition is Critical.**  Table 1 also reports the performance of datastores constructed from individual data sources. We find that certain sources greatly benefit specific benchmarks, e.g., Educational Text improves MMLU and GPQA; Math corpora boosts

Table 1: Results with Llama 3.1 8B Instruct, comparing no retrieval, single-source datastores, and COMPACTDS. $k = 3$, unless specified otherwise. Best results per source bolded in black, best overall bolded in blue. The relatives gains are computed using the best results with COMPACTDS for each dataset.

| | MMLU | | | | MMLU Pro | AGI Eval | MATH | GPQA | | | AVG |
|---|---|---|---|---|---|---|---|---|---|---|---|
| | STEM | Human. | Social | Others | | | | Phys | Bio | Chem | |
| No Retrieval | 60.2 | 72.0 | 78.7 | 68.9 | 39.8 | 56.2 | 46.9 | 26.7 | 47.4 | 25.7 | 48.3 |
| *Single-source datastores (All ANN only)* | | | | | | | | | | | |
| Math | **63.5** | 73.1 | 80.4 | 70.6 | 44.1 | **58.0** | **52.7** | 31.6 | 47.4 | 26.8 | **51.6** |
| Educational Text | 62.2 | **75.7** | **82.2** | **75.2** | **47.4** | 56.0 | 45.9 | **35.3** | 50.0 | 26.8 | 51.3 |
| High-quality CC | 62.3 | 74.0 | 82.8 | 74.3 | 45.1 | 56.8 | 45.9 | 26.7 | 46.2 | 26.2 | 50.0 |
| Books | 60.1 | 74.8 | 81.8 | 73.1 | 44.0 | 56.5 | 47.3 | 31.6 | 37.2 | 26.8 | 49.9 |
| Academic Papers (PeS2o) | 59.4 | 73.5 | 80.2 | 69.8 | 42.3 | 55.5 | 45.1 | 32.6 | 52.6 | **28.4** | 49.4 |
| Wikipedia (Redpajama) | 61.6 | 73.8 | 80.5 | 71.6 | 43.0 | 54.6 | 48.4 | 28.9 | 46.2 | 24.6 | 49.4 |
| Reddit | 60.6 | 72.8 | 78.9 | 70.6 | 41.4 | 56.5 | 45.9 | 31.0 | 50.0 | 24.6 | 49.0 |
| Wikipedia (DPR) | 61.4 | 74.1 | 80.8 | 71.1 | 41.9 | 55.6 | 46.1 | 25.1 | **53.8** | 26.8 | 49.0 |
| StackExchange | 63.0 | 72.0 | 78.5 | 69.5 | 41.0 | 57.0 | 46.9 | 31.0 | 46.2 | 21.9 | 48.9 |
| Github | 60.8 | 72.2 | 78.8 | 69.1 | 40.4 | 57.0 | 44.7 | 32.1 | 39.7 | 25.7 | 48.4 |
| PubMed | 60.4 | 72.6 | 79.6 | 70.1 | 40.7 | 56.1 | 44.9 | 28.9 | 47.4 | 25.1 | 48.4 |
| Academic Papers (ArXiv) | 59.3 | 71.8 | 78.1 | 69.6 | 39.5 | 57.5 | 45.7 | 25.7 | 38.5 | 27.9 | 48.0 |
| COMPACTDS-ANN only | 64.6 | 76.4 | **84.3** | 75.3 | 47.7 | 58.9 | 50.3 | 26.7 | 44.9 | 26.8 | 52.2 |
| COMPACTDS | 64.4 | 76.8 | 83.7 | 73.9 | 49.1 | **60.2** | 55.1 | **33.2** | 39.7 | 28.4 | 54.1 |
| COMPACTDS ($k = 10$) | **66.8** | **77.9** | 83.2 | **77.0** | **53.1** | 58.9 | **55.9** | 29.4 | 47.4 | **29.0** | **55.1** |
| *Relative gains from No Retrieval* | *11.0%* | *8.1%* | *7.1%* | *11.8%* | *33.4%* | *7.1%* | *19.2%* | *24.3%* | *0.0%* | *12.8%* | *14.1%* |

Table 2: MMLU results on COMPACTDS v. MASSIVEDS using Llama 3.1 8B Instruct. *ES* indicates "Exact Search." Best numbers in bold.

| | RAM | Method | Accuracy |
|---|---|---|---|
| No Retrieval | - | - | 68.9 |
| MASSIVEDS | 12.4TB | ES | 73.6 |
| COMPACTDS | 0.5TB | ANN | 75.2 |
| COMPACTDS | 0.5TB | ANN → ES | **75.3** |

Table 3: Oracle performance on COMPACTDS with $k_{oracle} = 100$ and $k = 3$. Best numbers in bold.

| | MMLU | MMLU Pro | AGI Eval | **AVG** |
|---|---|---|---|---|
| No Retrieval | 68.9 | 39.8 | 56.2 | 55.0 |
| *Llama 3.1 8B Instruct with CompactDS* | | | | |
| Our Best | 75.7 | 53.1 | 60.2 | 63.0 |
| **Oracle** | **85.6** | **63.1** | 65.0 | **71.2** |
| *Llama 3.3 70B Instruct* | | | | |
| No Retrieval | 81.5 | 57.8 | **71.1** | 70.1 |

performance on MATH; Wikipedia from DPR helps GPQA Biology; and Pes2o improves GPQA Chemistry. However, gains from any single source are generally limited.

On the other hand, COMPACTDS-ANN, which retrieves across all sources using only ANN search, yields a 8.1% average improvement, highlighting the importance of diverse data coverage for broad task performance. Our preliminary experiments showed that even removing four of the least impactful sources (ArXiv, Books, GitHub, and Reddit) reduces performance (e.g., by 1.8% on GPQA), suggesting that long-tail data diversity plays a role; see §C.1 for details.

Among single-source datastores, educational content, which is often absent in web-crawled corpora, and expert content like Math deliver the greatest improvement. Also note that Wikipedia from DPR is the most commonly used datastore in the RAG literature (Karpukhin et al., 2020; Lewis et al., 2020) however here, it offers little benefit on average, and even hurts performance on many datasets.

**First Practical Web-scale Datastore.** Table 2 compares COMPACTDS with MASSIVEDS (Shao et al., 2024),[0] which, to our knowledge, is the only open source web-scale datastore. MASSIVEDS uses exact inner product search over a 12.4TB vector database, requiring sequential shard-wise search and aggregation due to its size, leading to impractical latency for deployment. Building an ANN index at this scale is also prohibitively memory-intensive.

---

[0]For fair comparison, we took MMLU retrieval results from MASSIVEDS provided by the original paper and ran augmentation using Llama 3.1 8B Inst.

Table 4: Comparison among different retrieval pipelines, using Llama 3.1 8B Instruct and $K = 1,000$. *ES* indicates "Exact Search." The best results are shown in bold.

| | MMLU | | | | MMLU Pro | AGI Eval | MATH | GPQA | | | AVG |
|---|---|---|---|---|---|---|---|---|---|---|---|
| | STEM | Human. | Social | Others | | | | Phys | Bio | Chem | |
| No Retrieval | 60.2 | 72.0 | 78.7 | 68.9 | 39.8 | 56.2 | 46.9 | 26.7 | 47.4 | 25.7 | 48.3 |
| ANN Only (Contriever) | 66.4 | 76.7 | **85.2** | 76.7 | 50.1 | 57.6 | 53.3 | 31.6 | 48.7 | 27.3 | 53.8 |
| ANN (Contriever) + ES (Contriever) | 64.8 | 75.8 | 83.6 | 75.5 | 50.0 | **59.0** | 52.9 | **32.1** | **51.3** | 24.6 | 53.6 |
| ANN (Contriever) + ES (GRIT) | **66.8** | **77.9** | 83.2 | **77.0** | **53.1** | 58.9 | **55.9** | 29.4 | 47.4 | **29.0** | **55.1** |

Table 2 shows that COMPACTDS, built from heavily filtered web crawl data and a few additional sources (§3.1), already outperforms MASSIVEDS on MMLU using only ANN retrieval—while requiring just 4% of the RAM. With exact search, performance improves further, likely offsetting any loss from ANN. Overall, these results demonstrate that careful construction of datastore content combined with approximate-then-exact retrieval enables COMPACTDS to be the first compact, efficient, and deployable web-scale datastore.

Further ablations on the two-stage retrieval design and memory–performance tradeoffs, including a result showing a $4\times$ reduction in index size with only small performance drop, are provided in §C.

**Ablations on Exact Search.** Table 4 shows that COMPACTDS, with Exact Search using GRIT after ANN, consistently improves performance in nearly all cases. Exact Search using CONTRIEVER—the same model as ANN—does not improve over ANN-only significantly. This suggests that the use of a more expressive model accounts for the majority of the improvement, showing the benefit of the two-stage design described in §3.2.

A natural follow-up question is the comparison to ANN based on GRIT, which is significantly more expensive due to its 4,096-dimensional vectors, demanding substantially more RAM and disk space than Contriever-based ANN. §C.3 experiments with GRIT-based ANN on a small data subset, showing that the approximate-then-exact pipeline still outperforms GRIT ANN alone, likely because combining two complementary models helps.

**Upper-bounding COMPACTDS Performance.** To quantify the upperbound performance achievable from COMPACTDS, we report results using three *oracle* passages from a pool of 100 retrieved by COMPACTDS-ANN (§3 for details). As shown in Table 3, oracle reranking yields a significant gain, increasing the average improvement over no retrieval from 8.0 to 16.2, and even surpassing the 70B model. This suggests that COMPACTDS's utility could be significantly higher with better dense retrieval (returning oracle passages) or a stronger generator that can effectively leverage 100 passages without being misled by distractors.

**Qualitative Analysis.** Table 6 presents an example of passages retrieved from COMPACTDS for a given query. The query requests the ratio of hydrogen atoms in the second excited state at extreme atmospheric temperatures of Sirius. The retrieved passage contains a similar question and solution concerning the ratio in the first excited state, demonstrating COMPACTDS's ability to return passages relevant and helpful to the query. A more detailed qualitative analysis is given in §E.

**Extended Decontamination.** Although we already applied standard n-gram decontamination, it is still reasonable to ask whether the gains arise from residual contamination not captured by the n-gram method. To this end, we conducted a stricter post-hoc decontamination by prompting GPT-5-mini to identify and remove passages containing the test question and its answer. This can be seen as the most conservative approach: we qualitatively found many false positives but few false negatives. Results, detailed in §C.7, show a slight performance drop, but the overall gains remain largely the same, indicating that contamination is not the primary source of improvements. We also note that LMs may already have been exposed to these datasets during pretraining, so the presence of positives in retrieval does not introduce a new source of contamination specific to our method.

## 4.3 COMPACTDS IS EFFECTIVE ACROSS MODEL SIZES AND FAMILY

**Gains Persist at 70B Scale.** Table 5 shows results from applying COMPACTDS to Llama 3.3 70B Instruct, one of the strongest open-weight LLMs at the time of conducting experiments, and compares

Table 5: COMPACTDS results with Llama 3.3 70B Instruct, Mistral 7B Instruct, and Qwen3 8B with $k = 10$. Gains are consistent across different model sizes and families.

| | MMLU | | | | MMLU Pro | AGI Eval | MATH | GPQA | | | AVG |
|---|---|---|---|---|---|---|---|---|---|---|---|
| | STEM | Human. | Social | Others | | | | Phys | Bio | Chem | |
| **Llama 3.1 8B Instruct** | | | | | | | | | | | |
| No Retrieval | 60.2 | 72.0 | 78.7 | 68.9 | 39.8 | 56.2 | 46.9 | 26.7 | 47.4 | 25.7 | 48.3 |
| COMPACTDS | 66.8 | 77.9 | 83.2 | 77.0 | 53.1 | 58.9 | 55.9 | 29.4 | 47.4 | 29.0 | 55.1 |
| **Llama 3.3 70B Instruct** | | | | | | | | | | | |
| No Retrieval | 74.7 | 83.7 | 90.0 | 81.0 | 57.8 | 71.1 | 72.3 | 64.2 | 78.2 | 50.8 | 68.8 |
| COMPACTDS | 78.6 | 85.8 | 89.8 | 85.7 | 65.4 | 72.2 | 77.0 | 62.0 | 73.1 | 45.4 | 71.2 |
| **Mistral 7B Instruct** | | | | | | | | | | | |
| No Retrieval | 49.6 | 66.8 | 73.3 | 64.4 | 32.6 | 48.6 | 13.4 | 27.8 | 44.9 | 22.4 | 37.1 |
| COMPACTDS | 59.8 | 72.8 | 78.6 | 72.3 | 42.8 | 52.9 | 16.3 | 33.7 | 38.5 | 25.1 | 42.6 |
| **Qwen3 8B** | | | | | | | | | | | |
| No Retrieval | 73.2 | 75.2 | 82.8 | 75.6 | 49.4 | 65.1 | 56.3 | 38.5 | 56.4 | 29.5 | 57.0 |
| COMPACTDS | 79.5 | 80.5 | 87.1 | 81.9 | 60.6 | 67.3 | 60.3 | 42.2 | 52.6 | 27.3 | 61.6 |

them with Llama 3.1 8B Instruct. Significant gains are observed across most datasets, including 5% on MMLU STEM, 13% on MMLU Pro, and 7% on MATH. Smaller improvements on other MMLU subsets are likely due to performance saturation even without retrieval.

GPQA is the one exception where COMPACTDS offers no improvement at 70B, despite significant gains at 8B. This may be due to the dramatic performance gains of the no retrieval baseline from the 3.1 8B model to the 3.3 70B model (e.g., $26.7 \rightarrow 64.2$ on Physics), possibly due to the enhanced CoT capabilities. Again, adapting CoT to reranking may help, which we leave for future work.

**Significant Gains Across Different Models Families.** Table 5 shows results with two widely used LLMs outside the Llama family: Mistral 7B Instruct (Jiang et al., 2023) and Qwen3 8B (Yang et al., 2025). COMPACTDS improves performance consistently across datasets and model families, e.g., +10.2% on both MMLU STEM and MMLU Pro with Mistral, and +11.2% on MMLU Pro with Qwen3. GPQA Biology is an exception, likely for the same reason as in the 70B model.

Later, in §5.2, we further demonstrate that these gains persist with QwQ 32B, a strong reasoning model, matching or outperforming Search-o1 (Li et al., 2025b).

Table 6: Example of the top retrieved passage in GPQA.

| | |
|---|---|
| Query | `Question: Sirius is the brightest star in the sky. The temperature of this star is around 10000 K. Consider Hydrogen atoms in the atmosphere of Sirius. What is the ratio of the number of hydrogen atoms in the second excited state of Hydrogen to those in ground state?`
`Choices:(A) 8.2 * 10**-8 (B) 7.26 * 10^-6 (C) 8.11 * 10^-7 (D) 5.4 * 10**-9` |
| Retrieved | `...The temperature of the surface of a certain star is 8000 K. Most hydrogen atoms at the surface of the star are in the electronic ground state. What is the approximate fraction of the hydrogen atoms that are in the first excited state (and therefore could emit a photon)? The energy of the first excited state above the ground state is (-13.6/22 eV) - (-13.6 eV) = 10.2 eV = 1.632e-18 J. 2. Relevant equations Kb = 1.38e-23 J/K 1/T = E/Kb 3. The attempt at a solution I don't really know how to do this problem. My guess was to divide the energy in joules by the temperature and then divide by the boltzmann constant to make it unitless. (1.632e-18/8000)/(1.38e-23)=14.8....` |

## 5 COMPARISON TO SEARCH ENGINES AND AGENTIC SYSTEMS

Web search engines like Google and Bing are powerful retrieval models, and recent work that uses retrieval as an out-of-the-box tool often directly uses web search engines instead of in-house datastores (Li et al., 2025b; Wu et al., 2025). Prior work has rarely compared in-house datastores with web search engines, partly because many RAG benchmarks are constructed with search engines as oracles (e.g., Natural Questions (Kwiatkowski et al., 2019)), and these commercial engines have been optimized for decades. However, it is an open question whether search engines are necessarily

Table 7: Comparison between search engine and COMPACTDS, all with Llama 3.1 8B Instruct. The best results are bolded.

| | MMLU | | | | MMLU Pro | AGI Eval | MATH | GPQA | | | AVG |
|---|---|---|---|---|---|---|---|---|---|---|---|
| | STEM | Human. | Social | Others | | | | Phys | Bio | Chem | |
| No Retrieval | 60.2 | 72.0 | 78.7 | 68.9 | 39.8 | 56.2 | 46.9 | 26.7 | **47.4** | 25.7 | 48.3 |
| Search Engine | 61.8 | 72.6 | 80.0 | 71.6 | 42.8 | 59.7 | 51.4 | 25.7 | 46.2 | 32.2 | 51.3 |
| + LM Reranking | 61.3 | 73.5 | 80.3 | 72.0 | 44.0 | **59.8** | 50.2 | 32.1 | 42.3 | 29.5 | 51.5 |
| COMPACTDS | 66.8 | **77.9** | 83.2 | 77.0 | 53.1 | 58.9 | **55.9** | 29.4 | **47.4** | 29.0 | 55.1 |
| + LM Reranking | **69.1** | 77.8 | **86.8** | **78.7** | **54.6** | 59.5 | 53.0 | **33.7** | **47.4** | **33.3** | **56.0** |

optimal for LLMs, as they are not primarily designed for RAG. Furthermore, search engines present challenges for research: they are non-deterministic, costly, and often return noisy results.

In this section, we present a RAG pipeline using Google Search for retrieval (§5.1) and compare its performance with COMPACTDS (§5.2).

## 5.1 A COMPETITIVE METHOD FOR USING A SEARCH ENGINE

We construct RAG pipelines similar to that in §3.2, replacing COMPACTDS with a Google Programmable Search Engine[1] queried through the Custom Search API.

**Data Processing.** The search engine returns a ranked list of ten URLs, which include both web pages and PDFs. Extracting clean text from these sources is non-trivial. We broadly follow prior work using search engines (Li et al., 2025b;c), choosing methods to closely match the processing quality with COMPACTDS. For web pages, we parse the web page content using Resiliparse (Bevendorff et al., 2018), as in (Li et al., 2024), and fall back to BeautifulSoup (Richardson, 2007) when parsing fails, following (Li et al., 2025b;c). For PDFs, we use olmOCR (Poznanski et al., 2025), a vision-language model-based parser that outperforms traditional tools.[2]

We find many URLs encountering issues such as request errors, Captchas, or other access restrictions. We also tried crawl4AI (UncleCode, 2024) and paid services such as Jina AI but they did not lead to better performance (see §D.4).

Since our evaluation benchmarks are well-established, they appear frequently in web search results. To prevent contamination, we apply strict filtering by removing any paragraph (delimited by "\n\n") in a retrieved document that shares any 13-gram overlap with its corresponding query, following Shao et al. (2024), and also block search results from `huggingface.co`.

**Using Search Results in Context.** We augment an LLM with search results as in COMPACTDS, with one key difference. Web search results in PDFs are often too long to fit into the generator. While prior work truncates these texts (Li et al., 2025b), we find this approach suboptimal. Instead, we chunk each document to segments with up to $c$ words and rerank them using CONTRIEVER; this resembles COMPACTDS's retrieval, with ranking constrained to search results. We use $c = 512$ and $k = 3$.

We discuss the impact of decontamination, details regarding augmentation strategies, and ablations on augmentation choices and hyperparameters in §D.

## 5.2 RESULTS

All results comparing the search engine and our in-house datastore (COMPACTDS) are provided in Table 7 and Table 8.

**Search engine improves downstream performance.** The first block of Table 7 shows that using a search engine consistently improves task performance, yielding an average relative gain of 6%,

---

[1] https://developers.google.com/custom-search
[2] For instance, Search-o1 (Li et al., 2025b) relies on PDFPlumber, which works primarily on machine-generated PDFs.

and further minor improvements from LM reranking. Later results (Table 8) show that our pipeline outperforms the Search-o1 pipeline (Li et al., 2025b), demonstrating its competitiveness.

**COMPACTDS is a Competitive Alternative.** However, compared to the gains from COMPACTDS (second block of Table 7), web search is consistently outperformed. On average, COMPACTDS yields a 14% average relative improvement, whereas the search engine achieves only 6%. The gap is especially significant on certain datasets, e.g., MMLU Pro (44.0 vs. 54.6 after LM reranking). Such differences could not be observed in prior RAG benchmarks like Natural Questions, which was constructed using Google Search as oracle.

Later results (Table 8) suggest that relative performance may depend on the task and model, e.g., with QwQ, the search engine and in-house datastore perform comparably. Nonetheless, we found no case where COMPACTDS was meaningfully worse than the search engine.

**Experiments with QwQ 32B and Comparison to Search-o1.** We evaluate COMPACTDS with QwQ 32B (Team, 2024) and compare against Search-o1, an agentic RAG

Table 8: Experiments based on QwQ 32B, comparing our in-house datastore (COMPACTDS) and search engine pipelines from those in Search-o1 (Li et al., 2025b).

| | Self-contained? | GPQA-D | MATH-500 |
|---|---|---|---|
| *Li et al. (2025b)* | | | |
| No Retrieval | ✓ | 58.1 | 83.2 |
| RAG (w/ search engine) | ✗ | 61.6 | 85.0 |
| Search-o1 | ✗ | **63.6** | **86.4** |
| *Ours* | | | |
| No Retrieval | ✓ | 58.1 | 91.0 |
| RAG (w/ search engine) | ✗ | 63.1 | **94.0** |
| RAG (w/ COMPACTDS) | ✓ | 63.1 | 93.2 |

system built on web search and QwQ. This comparison serves three goals: (1) to validate our findings with a larger, competitive reasoning model; (2) to demonstrate the strength of our web search pipeline relative to prior work; and (3) to contrast minimal RAG with more complex agentic RAG systems. Evaluations are conducted on GPQA Diamond (Rein et al., 2023) and MATH-500 (Lightman et al., 2023), using the same benchmarks reported by Search-o1.

Table 8 highlights three key findings. First, our experimental setup is competitive: our no-retrieval performance matches or exceeds Search-o1's (e.g., outperforming on MATH-500 by 8%), and our web search pipeline achieves better results (e.g., 63.1 vs. 61.6 on GPQA Diamond). Second, minimal RAG with COMPACTDS remains strong, consistently improving over no-retrieval and matching our own web search results. Third, our minimal RAG with COMPACTDS matches or exceeds the full Search-o1 system. This suggest that a well-designed retrieval pipeline and a carefully constructed in-house datastore can serve as stronger baselines for future agentic RAG research. We additionally study how search engines and in-house retrieval can complement each other in §D.

## 6 CONCLUSION & DISCUSSION

We challenge the prevailing view that retrieval is ineffective for established, reasoning-intensive benchmarks. To do so, we introduce COMPACTDS, the first practical datastore that captures the diversity and scale of pre-training corpora while remaining deployable on a single node. Its core design combines a compact set of diverse sources with a two-stage approximate-then-exact retrieval system. Using COMPACTDS, a minimal RAG pipeline achieves consistent and significant gains across all datasets and model sizes, from 8B to 70B. COMPACTDS also outperforms agentic RAG pipelines based on Google Search, offering greater cost-efficiency and reproducibility.

Subsequent work (Liu et al., 2026) since the first release of COMPACTDS further improves the efficiency of the retrieval pipeline by applying DiskANN (Jayaram Subramanya et al., 2019). Furthermore, they publicly release a web demo and an API endpoint to further increase accessibility. Fang et al. (2025) also concurrently and independently demonstrates that using pretraining data as a retrieval corpora is an effective and efficient method to improve base model performance on tasks such as MMLU and MATH-500.

We release COMPACTDS to support future research in retrieval, reranking, and agentic RAG systems. For example, COMPACTDS could serve as a search engine within a local agentic framework for both inference and training, providing a strong alternative to commercial search APIs. It is also a realistic testbed for evaluating future reasoning-intensive retrieval and reranking pipelines at scale. Beyond retrieval-augmented generation, our dense retrieval pipeline can further enable efficient data attribution and contamination detection in pretraining data, which we leave for future work.

## 7 REPRODUCIBILITY STATEMENT

We describe the datastore curation, construction, and usage at a high-level in §3 and in greater detail in §A. Search engine methodology can be found in §5 and §D. Our experimental setting is detailed in §4 and §B. We also provide links to our project artifacts and codebase on the first page.

## ACKNOWLEDGEMENT

We thank Alex Fang, Yichuan Wang, Jinjian Liu, and Ai2 AllenNLP team members for valuable discussion and feedback.

SM was supported in part by a grant from DARPA to the Simons Institute for the Theory of Computing. PWK was supported by the Singapore National Research Foundation and the National AI Group in the Singapore Ministry of Digital Development and Information under the AI Visiting Professorship Programme (award number AIVP-2024-001), and by the AI2050 program at Schmidt Sciences.

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

# A    METHOD DETAILS

## A.1    DECONTAMINATION IN PREPROCESSING.

To ensure the robustness of our evaluation, we filter out paragraphs in the these data with an over 70% 13-gram Jaccard similarity with any query in all of our evaluation datasets following (Borgeaud et al., 2022). To quantify the effect of decontamination, we present the aggregated results from running RAG with eight single-source datastores before and after the decontamination in Figure 2. We observe a slight decline in performance on MMLU, MMLU Pro, and AGI Eval. Surprisingly, the performance improves slightly on GPQA and significantly on MATH. We suspect that this is due to two factors: (1) both GPQA and MATH are challenging tasks that involves intense

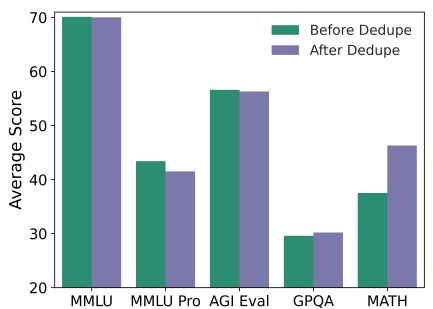

Figure 2: Effect of decontamination across eight single-source datastores.

reasoning and (2) duplicated n-grams in the retrieved passages might affect the chain-of-thought process negatively.

## A.2    COMPACTDS STATISTICS

Table 9 reports the number of passages, number of words, and number of chunks for each one of the twelve data sources. In total, COMPACTDS contains 639.2M passages with 380.5B words in total. We use a chunk size of 256 words following (Shao et al., 2024), resulting in 1.85B chunks.

Table 9:  Statistics of the data source for COMPACTDS.

|  | # Documents (M) | # Words (B) | # Passages (M) |
|---|---|---|---|
| Math | 6.4 | 7.8 | 33.7 |
| High-quality CC | 407.3 | 171.9 | 895.1 |
| Books | 0.2 | 7.8 | 69.5 |
| Academic Papers (PeS2o) | 7.8 | 49.7 | 198.1 |
| Wikipedia (Redpajama) | 29.8 | 10.8 | 60.5 |
| Reddit | 47.3 | 7.5 | 54.1 |
| Wikipedia (DPR) | 21.0 | 2.2 | 21.0 |
| Stack Exchange | 29.8 | 9.2 | 50.5 |
| Github | 28.8 | 17.1 | 84.3 |
| PubMed | 58.6 | 3.6 | 60.4 |
| Academic Papers (Arxiv) | 1.6 | 11.3 | 44.9 |
| Total | 639.2 | 380.5 | 1,851.8 |

**Index Hyperparameter Descriptions.**    We adopt FAISS (Johnson et al., 2019) to build the ANN index described in §3.2. The descriptions for each of IVFPQ hyperparameters are listed below:

*Number of Clusters* is the number of centroids that the vector space is partitioned into during nearest neighbor search. It positively correlates with index building speed and negatively correlates with the search time. A suggested value is $\sqrt{\#vectors}$.

*Number of Sub-quantizers* is the number of subspaces that the original vector is divided into for product quantization. It positively correlates with the performance, and is linear with the size of the resulting index. Some common values are $16, 32, 64$.

*Number of Training Samples* is the number of training samples used ot train the sub-quantizers and the product quantization centroid set. A suggested value is $0.05 \times \#vectors$.

*Number of probes.* the number of centroids to search for during search time. It positively correlates with the performance and the search time.

Table 10: Comparison between IVFPQ and Flat indices with Math and PeS2o. All results use Llama 3.1 8B Instruct with $k = 3$.

| | MMLU | | | | MMLU Pro | AGI Eval | MATH | GPQA | | | AVG |
|---|---|---|---|---|---|---|---|---|---|---|---|
| | STEM | Human. | Social | Others | | | | Phys | Bio | Chem | |
| No Retrieval | 60.2 | 72.0 | 78.7 | 68.9 | 39.8 | 56.2 | 46.9 | 26.7 | 47.4 | 25.7 | 48.3 |
| *Math* | | | | | | | | | | | |
| IVFPQ | 64.2 | 73.5 | 80.3 | 70.1 | 43.4 | 57.4 | 50.6 | 32.1 | 50.0 | 22.4 | 50.7 |
| Flat | 63.5 | 73.1 | 80.4 | 70.6 | 44.1 | 58.0 | 52.7 | 31.6 | 47.4 | 26.8 | 51.6 |
| *PeS2o* | | | | | | | | | | | |
| IVFPQ | 58.8 | 73.6 | 79.8 | 70.0 | 42.1 | 55.9 | 45.7 | 30.5 | 53.8 | 29.0 | 49.4 |
| Flat | 59.4 | 73.5 | 80.2 | 69.8 | 42.3 | 55.5 | 45.1 | 32.6 | 52.6 | 28.4 | 49.4 |

**Reproducibility and Index Type** As we use Inverted File Product Quantization (IVFPQ) index for approximating nearest neighbor search, potential discrepancies could exist among resulting indices from different training runs. To provide reference for reproducibility, we build Flat indices for Math and PeS2o that performs exact nearest neighbor search. We measure Recall@10 on GPQA for the IVFPQ indices using the retrieval results from Flat as the ground truth, and obtained 0.82 and 0.80. Table 10 shows the comparison between the performance of Flat and IVFPQ. We see that Flat outperforms IVFPQ by 0.9 on average with Math and performs similarly with IVFPQ with PeS2o. To exactly reproduce our results, the user should use Flat Index to be deterministic.

### A.3 LM Reranking

Dense retrieval can be limited as it is based on the inner product without cross-attention between the query and the passage. To mitigate this, we consider an optional reranking stage using an LLM (Sun et al., 2023) between dense retrieval and generation. The reranker $r$ assigns a relevance score to each passage $p$ given a query $q$. We do so by carefully designing a prompt that presents $q$ and $p$ to an LLM and instructs it to generate a helpfulness score (see details in §C.6). We then select the top-$k_{\text{rerank}}$ passages with the highest scores $\dot{p}_1, \dot{p}_2, \cdots, \dot{p}_{k_{\text{rerank}}} = \arg\text{Top}k_{\text{rerank}\,1 \leq j \leq K} \, r(\hat{p}_j | q)$, which then replace $\hat{p}_1, \hat{p}_2, \cdots \hat{p}_k$ in the generation stage. In our experiments, we make sure that the LM used for reranking is always identical to the generator. See Table 25 for the full prompt, and see the ablation using this method in Section C.4.

## B Evaluation Details

As described in §4.1. We evaluate COMPACTDS using the following benchmarks:

- **MMLU** (Hendrycks et al., 2021a) is a widely used benchmark for general knowledge and reasoning, consisting of 57 multiple-choice tasks spanning subjects from college-level math, physics, and computer science to professional-level law, medicine, and psychology. These tasks are grouped into four broad categories: STEM, Humanities, Social Sciences, and Other.

- **MMLU Pro** (Wang et al., 2024) is a challenging, reasoning-focused benchmarks across 14 disciplines, including math, physics, chemistry, business, and philosophy. Each question comes with ten answer choices.

- **AGI Eval** (Zhong et al., 2023) includes questions from human-centric standardized exams such as the SAT and LSAT. We evaluate the English multiple-choice tasks with five options each.

- **GPQA** (Rein et al., 2023) includes 448 expert-written, graduate-level multiple-choice questions in biology, physics, and chemistry, requiring deep understanding of topics like quantum mechanics, thermodynamics, and classical mechanics. GPQA is designed to be web-search-proof, being difficult for non-experts with unrestricted web access. We evaluate on GPQA using Chain-of-Thought (CoT).

- **MATH** (Hendrycks et al., 2021b) is a collection of challenging, competition mathematics problems, sourcing problems from exams such as AMC 12 and AIME. We evaluate using CoT.

- **GPQA Diamond** (Rein et al., 2023) is a high-quality subset of GPQA with 198 samples where most of the expert annotators answered correctly but most of the non-expert annotators answered incorrectly.
- **MATH-500** (Lightman et al., 2023) is a subset of MATH (Hendrycks et al., 2021b) with 500 samples randomly selected by OpenAI.

Table 11 shows the specific evaluation setups for the five main tasks, using OLMES (Gu et al., 2025) as our evaluation framework. For GPQA and MATH, we use the GPQA:0SHOT_COT::LLAMA3.1 and MINERVA_MATH::LLAMA3.1 configurations from OLMES, respectively, which reproduces the evaluation configuration used by Llama 3.1 in (Dubey et al., 2024). We use the MMLU:MC::OLMES, MMLU_PRO:MC::NONE, and AGI_EVAL_ENGLISH::OLMES OLMES configurations for MMLU, MMLU Pro, and AGI Eval, respectively.

For all benchmarks except GPQA, we randomly sample 100 questions per fine-grained category. Fine-grained categories for MMLU and MMLU Pro are defined in their respective works (Hendrycks et al., 2021a; Wang et al., 2024), while we use OLMES' selected categories for AGI-Eval and MATH. For GPQA, we evaluate on the entire GPQA **main set** (Rein et al., 2023).

For evaluation on GPQA **diamond set** and MATH-500 (both well-documented subsets of GPQA and MATH, respectively), we use the generation parameters from Search-o1 (Li et al., 2025b). Specifically, the generation settings are: a maximum of 32,768 tokens, temperature of 0.7, top_p of 0.8, a top_k of 20, and a repetition penalty of 1.05. However, for our simple retrieval, we use the same prompting style from the GPQA:0SHOT_COT::LLAMA3.1 and MINERVA_MATH::LLAMA3.1 configurations.

For MMLU and GPQA, we report micro-averages over the reported broad categories. We use the specific-category-to-broad-category map for MMLU STEM, Humanities, Social Sciences, and Other reported in (Hendrycks et al., 2021a). For GPQA, questions include metadata mapping them to their respective broad categories (Physics, Biology, Chemistry). For other benchmarks, we report an aggregate micro-average over the whole benchmark.

Table 11: Evaluation Setup for the five tasks evaluated with COMPACTDS.

|  | # Categories | # Samples | Type | Chat Format | # Demonstrations |
|---|---|---|---|---|---|
| MMLU (Hendrycks et al., 2021a) | 57 | 5,700 | MC | ✗ | 5 |
| MMLU PRO (Wang et al., 2024) | 14 | 1,400 | MC | ✗ | 5 |
| AGI Eval (Zhong et al., 2023) | 8 | 800 | MC | ✗ | 0 |
| GPQA (Rein et al., 2023) | 3 | 448 | COT | ✓ | 0 |
| MATH (Hendrycks et al., 2021b) | 7 | 700 | COT | ✓ | 0 |

## C  ADDITIONAL RESULTS ON COMPACTDS

### C.1  DATA SOURCE LEAVE-ONE-OUT

To further investigate the impact of individual data sources from §4.2, we conduct leave-one-out ablations where the retrieved passages from one data source are excluded at a time. Table 12 reports the complete results. For instance, MATH drops 2.6% when Math is excluded, MMLU Pro drops 1.6% when Educational Text is excluded, and GPQA Bio drops 3.9% when PubMed is excluded. This implies that the diversity of the data source is crucial. However, the performance can sometimes improve when excluding data sources, e.g., GPQA Bio improves by 5.1% after excluding High-quality CC, indicating that the retrieval can sometimes pick up noisy passages.

### C.2  IMPACT OF NUMBER OF RETRIEVED PASSAGES

Figure 3 shows the impact of varying the number of retrieved passages to feed into the generator ($k$). At $k = 10$, we observe the saturation in performance across the datasets. With GPQA and MATH—the two COT datasets—the performance declines significantly after $k = 10$. We find that this is due to the observed phenomenon that the model begins to generate blank outputs when provided with more retrieved passages.

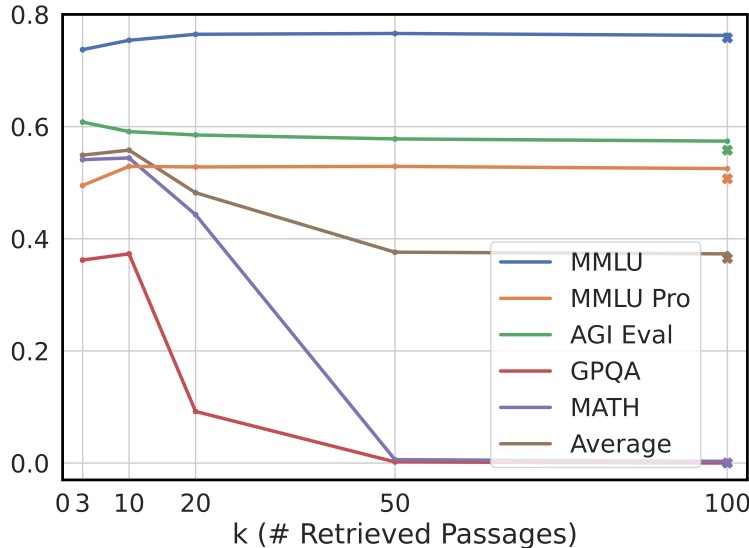

Figure 3: Effect of number of retrieved passages with COMPACTDS. 'X' represents the points where we apply the reordering method from (Jin et al., 2024) at $k = 100$.

Table 12: Leave-one-out ablation for each data source in COMPACTDS. Numbers that increase the most per dataset are bolded in black; numbers that drop the most are bolded in blue.

| | MMLU | | | | MMLU Pro | AGI Eval | MATH | GPQA | | | AVG |
|---|---|---|---|---|---|---|---|---|---|---|---|
| | STEM | Human. | Social | Others | | | | Phys | Bio | Chem | |
| No Retrieval | 60.2 | 72.0 | 78.7 | 68.9 | 39.8 | 56.2 | 46.9 | 26.7 | 47.4 | 25.7 | 48.3 |
| COMPACTDS-ANN only ($k = 3$) | 64.6 | 76.4 | 84.3 | 75.3 | 47.7 | 58.9 | 50.3 | 26.7 | 44.9 | 26.8 | 52.2 |
| *Leave-one-out (ANN only) ($k = 3$)* | | | | | | | | | | | |
| Math | **64.0** | **76.6** | 84.1 | 75.3 | 47.2 | **57.8** | **47.7** | **25.1** | 44.9 | **25.1** | **51.0** |
| Educational Text | **65.0** | **75.0** | **83.0** | **73.6** | **46.1** | 58.1 | 50.9 | 27.3 | 43.6 | **25.1** | 51.5 |
| High-quality CC | 64.4 | 75.7 | 84.0 | 74.2 | 47.6 | 58.2 | 49.3 | 27.8 | 43.6 | 26.8 | 51.8 |
| PubMed | 64.6 | 76.4 | 84.3 | 75.2 | 47.9 | 58.9 | 50.3 | 26.7 | **41.0** | 26.8 | 52.1 |
| StackExchange | 64.1 | 76.5 | 84.2 | 75.3 | 47.4 | **59.0** | 50.6 | 26.2 | 44.9 | 26.8 | 52.1 |
| Reddit | 64.9 | 75.7 | 84.2 | 74.9 | 47.6 | 58.8 | 50.6 | 26.7 | 44.9 | 26.8 | 52.1 |
| Books | 64.4 | 75.9 | 84.2 | 75.2 | 47.5 | 58.8 | 50.3 | 26.7 | 46.2 | 27.3 | 52.2 |
| Wikipedia (DPR) | 64.5 | 76.5 | 84.3 | 75.2 | **47.9** | 58.9 | 50.3 | 26.2 | 44.9 | 26.8 | 52.2 |
| Wikipedia (Redpajama) | 64.7 | 76.4 | 84.3 | 75.2 | 47.7 | 59.0 | 50.1 | 26.7 | 44.9 | 26.8 | 52.2 |
| Github | 64.7 | 76.4 | 84.3 | 75.3 | 47.7 | 58.9 | 50.3 | 26.7 | 44.9 | 26.8 | 52.2 |
| Academic Papers (ArXiv) | 64.5 | 76.3 | 84.3 | 75.3 | 47.7 | 58.8 | 50.4 | 30.5 | 44.9 | 26.8 | 52.5 |
| Academic Papers (PeS2o) | 64.6 | 76.0 | 84.2 | **75.7** | 47.8 | 58.9 | **50.9** | **31.0** | **50.0** | **28.4** | **53.0** |
| COMPACTDS ($k = 10$) | 66.8 | 77.9 | 83.2 | 77.0 | 53.1 | 58.9 | 55.9 | 29.4 | 47.4 | 29.0 | 55.1 |

Furthermore, we explored the reordering technique at $k = 100$ from (Jin et al., 2024) designed to improve the performance with long-context retrieval, where the most relevant passages are put at the beginning and end of the list of retrieved passages and the most irrelevant ones are hidden in the middle of the list. However, we find this technique ineffective in our setting according to Figure 3.

## C.3 EFFECT OF TWO-STAGE PIPELINE IN COMPACTDS

As discussed in §3, we use CONTRIEVER-MSMARCO (Izacard et al., 2021) for ANN and GRITLM-7B (Muennighoff et al., 2024) for exact nearest neighbor search. We experiment with different permutations of these two models over the two stages of the pipeline with 1% of COMPACTDS. Results are reported in Table 13. GRIT, as the larger model, results in higher ANN-only performance than CONTRIEVER. Using Exact Search with CONTRIEVER does not improve or even decreases the performance. Surprisingly, ANN with Contriever + Exact Search with GRIT outperforms using

Table 13: Comparison between Contriever and GritLM-7B for ANN and Exact Search on 1% of COMPACTDS with $k = 3$. *ES* stands for Exact Search. The best numbers are shown in bold.

| | MMLU | | | | MMLU Pro | AGI Eval | MATH | GPQA | | | AVG |
|---|---|---|---|---|---|---|---|---|---|---|---|
| | STEM | Human. | Social | Others | | | | Phys | Bio | Chem | |
| No Retrieval | 60.2 | 72.0 | 78.7 | 68.9 | 39.8 | 56.2 | 46.9 | 26.7 | 47.4 | 25.7 | 48.3 |
| ANN only with Contriever | 60.5 | 73.8 | 79.8 | 71.0 | **43.9** | 56.6 | 44.7 | 32.6 | 42.3 | 25.7 | 49.4 |
| +ES with Contriever | 60.3 | 73.4 | **80.4** | 70.5 | 42.6 | 56.9 | 46.4 | 24.6 | 47.4 | 30.1 | 49.4 |
| +ES with GritLM-7B | **61.1** | **74.6** | 79.5 | **71.5** | 42.8 | 56.2 | **47.1** | 30.5 | **59.0** | **34.4** | **50.8** |
| ANN only with GritLM-7B | 59.9 | 72.2 | 79.1 | 71.5 | 41.4 | **57.1** | 47.0 | 32.6 | 47.4 | 30.1 | 49.9 |
| +ES with Contriever | 61.2 | 73.2 | 80.2 | 71.0 | 43.0 | 56.6 | 47.1 | 29.4 | 42.3 | 25.7 | 49.5 |
| +ES with GritLM-7B | 59.8 | 73.7 | 79.8 | 71.1 | 42.9 | 56.5 | 45.7 | **35.8** | 52.6 | 29.0 | 50.2 |

Table 14: Comparison between two levels of approximation for ANN with COMPACTDS. The results are at $k = 10$. The best results are shown in bold.

| | Index Size | Performance | | | | | |
|---|---|---|---|---|---|---|---|
| | | MMLU | MMLU Pro | AGI Eval | MATH | GPQA | **AVG** |
| No Retrieval | - | 68.9 | 39.8 | 56.2 | 46.9 | 29.9 | 48.3 |
| **# Subquantizers = 64** | | | | | | | |
| COMPACTDS-ANN Only | 125GB | 74.7 | 50.0 | 59.1 | 50.6 | 32.6 | 53.4 |
| COMPACTDS | | 74.4 | 51.7 | **59.2** | 54.6 | 30.8 | 54.1 |
| **# Subquantizers = 256** | | | | | | | |
| COMPACTDS-ANN Only | 456GB | 75.2 | 50.1 | 57.6 | 53.3 | **32.8** | 53.8 |
| COMPACTDS | | **75.3** | **53.1** | 58.9 | **55.9** | 32.4 | **55.1** |

GRIT for both ANN and Exact Search. We hypothesize that this is because two different embedding models complement each other.

## C.4 IMPACT OF RERANKING

Table 15 quantifies the impact of LM reranking described in §3.2. LM reranking further improves several datasets, particularly MMLU and MMLU Pro, but struggles with MATH and GPQA. We hypothesize that LM reranking demands tailored reranking instructions for each downstream task: for instance, MATH and GPQA use CoT during generation, where reranking does not, and using CoT for reranking could further improve performance. Another possible factor is the limited capacity of our generator (Llama 3.1 8B Inst).

## C.5 4× SMALLER INDEX CAN BE ACHIEVED WITH 1% PERFORMANCE DROP

To assess the effect of approximation in ANN, we build two versions of COMPACTDS by varying the number of subquantizers in IVFPQ. The more compressed version (125GB) trades accuracy for memory, while the less compressed one (456GB) offers higher fidelity. As shown in Table 14, 4× compression reduces index size by a factor of 4, with only a 1% drop in average performance. We use the 456GB index as the default in this paper, but the 125GB index still preserves most gains over no retrieval, and this tradeoff can be adjusted based on available memory.

## C.6 ORACLE RERANKING IMPLEMENTATION

We formalize the oracle reranking method discussed in §4.2. Given a test query $q$, the ground truth answer a, and $K$ candidate passages from COMPACTDS-ANN, the oracle passages are

$$p_1^*, p_2^*, \cdots, p_{k_{\text{rerank}}}^* = \arg\text{Top}k_{\text{rerank}1 \leq j \leq K}(P_{\text{LM}}(a|p_j, q) - P_{\text{LM}}(a|q)).$$

where $P_{\text{LM}}(a)$ represents the probability that the LM assign to the gold answer $a$. These oracle passages then replace $\hat{p}_1, \hat{p}_2, \cdots \hat{p}_k$ in the generation stage.

Table 15: Comparison among different retrieval pipelines, using Llama 3.1 8B Instruct and $K = 1,000$. *ES* indicates "Exact Search." LM reranking uses $k_{\text{rerank}} = 100$. The best results are shown in bold.

| | MMLU | | | | MMLU Pro | AGI Eval | MATH | GPQA | | | AVG |
|---|---|---|---|---|---|---|---|---|---|---|---|
| | STEM | Human. | Social | Others | | | | Phys | Bio | Chem | |
| No Retrieval | 60.2 | 72.0 | 78.7 | 68.9 | 39.8 | 56.2 | 46.9 | 26.7 | 47.4 | 25.7 | 48.3 |
| ANN (Contriever) + ES (GRIT) | 66.8 | **77.9** | 83.2 | 77.0 | 53.1 | 58.9 | **55.9** | 29.4 | 47.4 | 29.0 | 55.1 |
| + LM Reranking | **69.1** | 77.8 | **86.8** | **78.7** | **54.6** | **59.5** | 53.0 | **33.7** | 47.4 | **33.3** | **56.0** |

Table 16: Deduplication with GPT-5 Mini on COMPACTDS with $K = 1000$, $k = 10$, and Llama 3.1 8B Instruct.

| | MMLU Pro | MATH | GPQA |
|---|---|---|---|
| No Retrieval | 39.8 | 46.9 | 29.9 |
| Before Dedup | 53.1 | 55.9 | 32.4 |
| *Relative gains from No Retrieval* | *33.4%* | *19.2%* | *8.4%* |
| After Dedup | 48.0 | 48.0 | 33.9 |
| *Relative gains from No Retrieval* | *20.6%* | *2.2%* | *13.4%* |

## C.7 A LOWERBOUND WITH LM DEDUPLICATION

To consider potential duplicated text with minor format difference with the query, which would be neglected by the 13-gram filtering, we adpot GPT-5 Mini (OpenAI, 2025) to identify retrieved passages by COMPACTDS (See Table 27 for our contamination detection instruction). Despite instructing the model to only identify the passages that contain the exact same question and answer, the model still marks the passages that contain either the question and multiple candidate answers without identifying the correct one or information that can be used to infer about the answer to the question (e.g., as shown in Table 17 and Table 18). As a result, we consider this decontamination to be overstrict and treat it as a lowerbound of COMPACTDS's performance. We conduct such deduplication on three datasets that are more likely to contain deuplicates, and the results are shown in Table 16.

Table 17: Example of a False Positive from MMLU Pro in LM Deduplication.

| | |
|---|---|
| Query/Answer | The engineer Camillo Oliver was 40 years old when he started the company in 1908. At his factory in Ivrea, he designed and produced the first Italian typewriter. Today the company's head office s still in Ivrea, near Turin, but the company is much larger than it was in those days and there are offices all around the world.By 1930 there was a staff of 700 and the company turned out 13,000 machines a year. Some went to customers in Italy, but Olivetti exported more typewriters to other countries.Camillo's son, Adriano, started working for the company in 1924 and later he became the boss. He introduced a standard speed for the production line and he employed technology and design specialists. The company developed new and better typewriters and then calculators(\u8ba1\u7b97\u673a). In 1959 it produced the ELEA computer system. This was the first mainframe\uff08\u4e3b\u673a\uff09computer designed and made in Italy.After Adriano died in 1960, the company had a period of financial problems. Other companies, especially the Japanese, made faster progress in electronic technology than the Italian company. In 1978, Carlo de Benedetti became the new boss. Olivetti increased its marking and service networks and made agreements with other companies to design and produce more advanced office equipment. Soon it became one of the world's leading companies in information technology and communications. There are now five independent companies in the Olivetti group\u2014one for personal computers, one for Systems and services, and two for telecommunications.\nQuestion: From the text we learn that ______________.\n A. by 1930 Olivetti produced 13,000 typewriters a year\n B. Olivetti earned more in the 1960s than in the 1950s\n C. some of Olivetti\u2019s 700 staff regularly visited customers in Italy\n D. Olivetti set up offices in other countries from the very beginning \nAnswer: A |
| Retrieved | the most successful and prosperous companies in the quite immature Italian industrial scenario. The first factory in Ivrea was enlarged, floor space was added and a considerable number of new employees were hired; between 1914 and 1929, the production of typewriters rose 10 times from 1,300 to 13,000. New typewriter models were added to the line, such as the M20 in 1920 and the M40 10 years later, and Olivetti started international expansion, opening sales offices in six foreign countries. Fig. 4.1 The olivetti M1. Source: www.olivetti.it In 1924, Adriano Olivetti started working in the family business as an apprentice. One year later, together with Domenico Burzio, he travelled around the USA and visited many factories. This tour opened his mind and offered him a range of new ideas he decided to put in practice, as he returned to Italy, to modernize Olivetti, through innovative and advanced projects: function-based management, decentralized staff organization, rationalization of assembly work, development of an efficient national and international sales network. The new organization produced very positive effects, increasing productivity and sales. In 1931, Adriano travelled to the Soviet Union with a delegation of Italian industrialists; later that year, he established an advertising department in Olivetti, which started working straightaway with famous artists and designers and he eventually set up the organization office. During this period, Olivetti was active on the social front, interested in the welfare and trying to improve the living conditions of its workers. In 1926, the company started building houses for its workmen and managed to set |

Table 18: Example of a False Positive from MATH in LM Deduplication.

| | |
|---|---|
| Query/Answer | `Problem:\nIn how many ways can 8 people be seated around a square table with 2 people on a side?  (Two configurations are considered equivalent if one is a rotation of another.)\n\nSolution: There are $8!$ ways to place the people around the table, but this counts each valid arrangement 4 times (if you move each person 2, 4, or 6 places clockwise you get the same arrangement).  The answer is $\\dfrac{8!}{4} = \\boxed{10080}$.` |
| Retrieved | `Q: Arrangements around a square, People not sit across diagonally – Extension The first two parts of this question are exact same as the Counting the arrangements of 8 people around a square table? and have already been answered but my struggle is in the final part of my problem which is an extension of the problem cited. Basically, the final part of the question ask how to arrange 8 people around a square with 2 people on each side such that 2 people who hate each other won't sit on the same side or on parallel side, i.e, they cannot sit across directly or diagonally. I solved the first two parts, where part a my answer was $$7!\\times2$$ and for the second part my answer was $$2\\times(7!-6!)$$ the second answer might be wrong but I am confident about the answer to the first part A: In how many ways can eight people be seated at a square table if there are two people per side up to rotations by $90^\\circ$, $180^\\circ$, $270^\\circ$, or $360^\\circ$? Your answer is correct. Ann can be seated in eight ways. As we proceed clockwise around the table from Ann, the others can be seated in $7!$ ways. Dividing by four to account for equivalent rotations yields $$\\frac{8 \\cdot 7!}{4} = 2 \\cdot 7!$$ as you found. In how many ways can eight people be seated at a square table if Ann and Bob do not sit on the same side? Again, your answer is correct. Ann can be seated in` |

# D  ADDITIONAL RESULTS ON SEARCH ENGINES

## D.1  SEARCH ENGINE AUGMENTATION METHOD DETAILS

We set up a default Google Custom Search Engine for web retrieval (except for blocking results from the huggingface.co domain). The search engine returns 10 results (one page) per query, with a paid daily limit of 10,000 queries at a rate of $5 per 1000 queries. Similar restrictions/costs exist for other search engine providers like Bing.

We limit retrieval to 10 results per query due to the aforementioned costs and rate-limiting.

**Ranking Details.**   As with COMPACTDS, we use CONTRIEVER-MSMARCO (Izacard et al., 2022), referred to as CONTRIEVER, as the default chunk embedder. GRITLM-7B (Muennighoff et al., 2024) is used for optional second-stage reranking (generally referred to as GRIT-reranking).

**Strategies to use Search Results in Context.**   We explore two methods for using web search results in context:

(1) **Rank→Extract** identifies the most relevant chunk per document using CONTRIEVER (Izacard et al., 2022) reranking, preserving search result order over the selected chunks.
(2) **Aggregate→Rank** aggregates all chunks across search results and reranks them using CONTRIEVER; this resembles COMPACTDS's retrieval, with ranking constrained to search results.

As some test queries were overly long or complex and caused search failures, we also consider **Break-down**, which decomposes the query into up to three subqueries, performs a search for each, and aggregates results (following the Aggregate→Rank strategy). Table 26 shows the prompt used for the **break-down** method to decompose queries into at most three subqueries prior to search engine retrieval.

In §5, we use **Aggregate→Rank** as our main strategy. Ablations over the different strategies are available in §D.3.

## D.2  IMPACT OF DECONTAMINATION ON SEARCH ENGINE RESULTS

As described in §5.1, to prevent contamination, we apply strict filtering by removing any paragraph (delimited by "\n\n") in a retrieved document that shares any 13-gram overlap with its corresponding query, following Shao et al. (2024), and also block search results from huggingface.co. Table 19

Table 19: Impact of Decontamination of Search Engine Retrieval Results. We bold the best results between before and after decontamination.

| | MMLU | MMLU Pro | GPQA |
|---|---|---|---|
| Our pipeline (Decon.) | 70.5 | 42.8 | 31.9 |
| Before Decon. | **73.1** | **46.4** | **33.3** |

Table 20: Impact of data sources in search results. Note that the first row is equivalent to *No Retrieval*.

| Web | PDFs | MMLU | MMLU Pro | AGI Eval | GPQA |
|---|---|---|---|---|---|
| ✗ | ✗ | 68.9 | 39.8 | 56.2 | 29.9 |
| ✗ | ✓ | 69.3 | 42.4 | 56.0 | **34.6** |
| ✓ | ✗ | 70.4 | 41.0 | **60.1** | 31.9 |
| ✓ | ✓ | **70.5** | **42.8** | 59.7 | 31.9 |

Table 21: Results of methods using Google Search for retrieval augmentation, using Llama 3.1 8B Instruct. The best results are bolded.

| | MMLU | | | | MMLU Pro | AGI Eval | MATH | GPQA | | | AVG |
|---|---|---|---|---|---|---|---|---|---|---|---|
| | STEM | Human. | Social | Others | | | | Phys | Bio | Chem | |
| No Retrieval | 60.2 | 72.0 | 78.7 | 68.9 | 39.8 | 56.2 | 46.9 | 26.7 | **47.4** | 25.7 | 48.3 |
| Rank→Extract | 62.4 | 72.8 | 79.3 | 71.4 | 42.9 | 59.4 | **51.6** | 29.9 | 43.6 | 26.8 | 51.1 |
| Aggregate→Rank | 61.8 | 72.6 | 80.0 | 71.6 | 42.8 | **59.7** | 51.4 | 25.7 | 46.2 | **32.2** | 51.3 |
| Break-down | **72.5** | **75.5** | **81.8** | **73.8** | **44.8** | 58.4 | 50.4 | **30.5** | 38.5 | 31.1 | **51.6** |

quantifies the impact of decontamination, showing that it can reduce downstream performance by up to 3.6%. This underscores the importance of rigorous decontamination when using search engines for evaluation.

### D.3 ADDITIONAL ABLATIONS OVER SEARCH ENGINE METHOD

We first compare the different strategies outlined in §D.1 to prepare search results for in-context use. Table 21 shows that incorporating search results consistently improves accuracy. *Aggregate→Rank* outperforms *Rank→Extract*, suggesting that ranking from search engines is not necessarily better than those from CONTRIEVER. *Break-down* performs slightly better overall, particularly on MMLU STEM, MMLU Pro, and GPQA Physics. However, due to *break-down* costing substantially more and only yielding marginal performance gains on average, we adopt *Aggregate→Rank* as our default method. We suspect that query decomposition may require dataset-specific tailoring and stronger decomposition models for more pronounced gains.

We conduct a brief investigation over chunk size $c$ and the number of chunks to feed into the generator $k$. Table 22 suggests that a larger chunk size generally yields slightly better performance, while the optimal $k$ varies by task. As the variants for $(c, k)$ yield similar performances, we default to using $(c = 512, k = 3)$ due to it being the cheapest hyperparameter configuration (i.e., smaller number of chunks to embed and shorter prompt length).

We also investigate the impact of GRIT- and LM-reranking over the top $K = 100$ CONTRIEVER-ranked chunks. Table 22 indicates that, unlike with COMPACTDS, GRIT-based reranking on web retrieval results generally yields similar or slightly worse performance than the baseline Aggregate→Rank strategy with CONTRIEVER. However, we do observe gains on GPQA Biology (46.2%→48.7%). LM reranking improves performance on tasks such as MMLU Pro and GPQA Physics, but leads to degradation on others, such as MATH and GPQA Biology. Similar to COMPACTDS, this likely demonstrates the weakness of a smaller reranker on more reasoning-intensive tasks, especially without enough CoT generation before scoring.

In general, we suspect that reranking may not see as much benefit in the case of web retrieval because of the decreased breadth in retrieval results. Although there may be more than 100 chunks per query to rerank, these chunks come from at most 10 documents; chunks retrieved from COMPACTDS don't necessarily adhere to such restrictions. The performance of oracle reranking further supports this hypothesis, as we see the approximate upperbound gain in Table 22 is considerably less than that seen by COMPACTDS with oracle reranking in Table 7.

Table 22: Results of different $(c, k)$ hyperparameter choices and reranking methods over Google Search retrieval results with Llama 3.1 8B Instruct. $c$ indicates the chunk size, and $k$ indicates the number of chunks fed into the generator. We use the **Aggregate→Rank** method for all variations. For reranking strategies, we rerank the top $K = 100$ CONTRIEVER-ranked chunks. The best results across $c, k$ hyperparameters and the ablations over reranking strategies, respectively, are bolded (excluding oracle).

| | MMLU | | | | MMLU Pro | AGI Eval | MATH | GPQA | | | AVG |
|---|---|---|---|---|---|---|---|---|---|---|---|
| | STEM | Human. | Social | Others | | | | Phys | Bio | Chem | |
| No Retrieval | 60.2 | 72.0 | 78.7 | 68.9 | 39.8 | 56.2 | 46.9 | 26.7 | 47.4 | 25.7 | 48.3 |
| *Ablations over $(c, k)$ Hyperparameters* | | | | | | | | | | | |
| $c = 256, k = 3$ | 60.9 | 72.5 | 79.3 | 71.5 | 42.6 | 58.3 | 50.2 | 27.8 | 41.0 | 28.4 | 50.3 |
| $c = 512, k = 3$ | 61.8 | 72.6 | 80.0 | 71.6 | 42.8 | **59.7** | 51.4 | 25.7 | **46.2** | **32.2** | **51.3** |
| $c = 512, k = 10$ | **62.8** | **72.7** | **80.1** | **71.7** | **43.4** | 59.3 | **51.8** | 26.7 | 46.2 | 29.5 | **51.3** |
| *Reranking Strategies with $(c = 512, k = 3)$* | | | | | | | | | | | |
| Aggregate→Rank | **61.8** | 72.6 | 80.0 | 71.6 | 42.8 | 59.7 | **51.4** | 25.7 | 46.2 | **32.2** | 51.3 |
| + GRIT Reranking | 61.1 | 72.5 | 79.5 | 70.9 | 42.8 | 59.5 | 51.1 | 27.8 | **48.7** | 29.5 | 51.1 |
| + LM Reranking | 61.3 | **73.5** | **80.3** | **72.0** | **44.0** | **59.8** | 50.2 | **32.1** | 42.3 | 29.5 | 51.5 |
| + Oracle Reranking | 65.9 | 77.0 | 84.2 | 76.6 | 48.2 | 61.5 | – | – | – | – | – |

Table 23: Comparison between different webpage URL parsing strategies with ($c = 512, k = 3$) and using the **Aggregate→Rank** strategy. **Static** parsing is the default method described in §5.1. The best results are in bold.

| | MMLU Pro | MATH | GPQA |
|---|---|---|---|
| No Retrieval | 39.8 | 46.9 | 29.9 |
| Static (Ours) | 42.8 | **51.6** | 31.9 |
| Crawl4AI | 41.9 | 46.7 | **33.7** |
| JINA | **43.4** | 48.6 | 32.4 |

Table 24: Comparison between using COMPACTDS only and merging COMPACTDS and search engine retrieval results at $k = 10$. The first row is equivalent to using no retrieval. Best results are in bold.

| COMPACTDS | Search Engine | MMLU Pro | MATH | GPQA |
|---|---|---|---|---|
| ✗ | ✗ | 39.8 | 46.9 | 29.9 |
| ✓ | ✗ | 53.1 | 55.9 | 32.4 |
| ✓ | ✓ | **53.5** | **56.4** | **33.0** |

## D.4 SEARCH ENGINE PIPELINE RESULTS WITH DYNAMIC URL PARSING

We perform ablations using dynamic URL parsing methods as an alternative to the static scraping presented in §5.1. We explore two options: 1) Crawl4AI (UncleCode, 2024) and 2) JINA Reader API[3]. Both frameworks offer high-quality URL content parsing by rendering pages within a browser and then formatting the data into Markdown format. In addition to higher-quality, more complete parsing, the dynamic page rendering with browser sessions allows for increased chances to avoid bot detection such as Captchas. Crawl4AI is a free, open-source web-scraping framework, while JINA Reader is a paid service charging on number of output tokens[4] that is simple to set-up (e.g., prepend `https://r.jina.ai/` to URLs before performing GET requests). We only replace the static parsing of webpage URLs and continue using olmOCR for PDFs because 1) Crawl4AI has more limited PDF parsing than olmOCR and 2) JINA Reader would be much more expensive due to the longer PDFs resulting in a far greater amount of output tokens.

After gathering the parsed results, we additionally filter out any embedded URLs (e.g., the Markdown conversion preserves URLs on the page) (following (Li et al., 2025b)) and parsed texts that contain keywords such as "cloudflare" or "captcha" indicating a bot detection page was parsed instead of the protected content. We then follow the rest of the pipeline detailed in §5.1, using the Aggregate→Rank method.

Table 23 demonstrates that the best URL parsing strategy varies across benchmark, but no single method consistently outperforms the others, not even the paid service (JINA). We suspect one possibility could be that the parsing quality, while non-trivial, is not as impactful as the significant

---

[3] `https://jina.ai/reader/`
[4] There is a free option that avoids an API key, but it is heavily rate-limited at 20 requests per minute.

amount of unparsable URLs due to preventative measures such as bot detection (e.g., as employed on `quizlet.com`), which greatly reduces the number of retrieval candidates. We note that even if one were to deploy stronger web-scraping services that could bypass these protections, there are still ethical concerns regarding non-permissive data usage that should be approached with caution.

For our experiments, we use static parsing as it is the fastest method while also not demonstrating significant performance degradation.

### D.5 COMPLEMENTARY STRENGTHS OF SEARCH AND IN-HOUSE RETRIEVAL.

While we demonstrated that in-house retrieval is a competitive alternative to search engines in §5.2, we believe that search engines and in-house retrieval offer complementary benefits. COMPACTDS is simple (as they rely solely on vector similarity), self-contained, reproducible, and robust; it avoids the cost and noise associated with web search (see §5.1). In contrast, search engines incorporate complex rule-based and ranking algorithms that complement vector-based retrieval. They also access a broader and more diverse set of documents, including content unavailable to crawlers, such as many web-hosted PDFs.

PDFs, in particular, are a notable advantage of web search. While COMPACTDS includes PDFs, they are primarily academic papers and educational text. In contrast, the web contains many valuable PDFs, such as lecture notes and problem set solutions, that are largely absent from COMPACTDS. To assess the impact of different source types, we compare performance using only web pages vs. only PDFs (Table 20). PDF-only retrieval can significantly improve performance on some tasks, e.g., on MMLU Pro, using only PDFs performs nearly as well as using both PDFs and web pages, and for GPQA, PDFs alone even outperform the combined. Overall, we think:

1. PDFs are a valuable source of information complementary to web crawl, similar to how educational text and academic papers were effective in §4.2.

2. Common Crawl does not represent the full scope of web content; expanding COMPACTDS with additional sources such as high-quality PDFs on the web found by search engines could enhance its performance.

### D.6 IMPROVING COMPACTDS WITH SEARCH ENGINES

As search engines naturally retrieve over a larger data distribution (e.g., the entire indexed web), we hypothesize that retrieved search results could augment COMPACTDS performance in a hybrid search fashion.

As a preliminary hybrid search strategy, we aggregate the top-$K$ results from both COMPACTDS and search engine retrieval, re-ordering them using their GRIT-based retrieval scores. We compare this to just GRIT-reranking the top $K = 1000$ results from COMPACTDS only. For a fair comparison (only reranking 1000 total candidates), we select the top $K_{\text{local}} = 900$ results from COMPACTDS and top $K_{\text{web}} = 100$ results from our search engine. Table 24 shows our merging strategy yields small gains on all benchmarks (53.1%→53.5% for MMLU Pro; 55.9%→56.4% for MATH; and 32.4%→33.0% for GPQA, respectively).

Our preliminary merging method is likely not optimal. However, we emphasize that our qualitative observations on the domain diversity of search engine results motivate further exploration into how search engines can complement local datastores outside of treating them as independent retrieval flows. For example, to expand COMPACTDS to specialized benchmarks in other domains (e.g., law) while also maintaining space efficiency, identifying a concise, relevant, and high-quality subset of data to build on efficiently is a natural first step. One could use search engine results over a benchmark validation subset (potentially using agentic search flows to improve search quality) to identify relevant URLs. Metadata from these URLs (e.g., domain, subfolders) could then be used as a pre-processing filter to reduce web crawl dumps to a much smaller, higher-quality subset (similar to how we currently use FineWeb-Edu filtering), or help with targeted crawls to collect data not typically found in pre-training data such as the web PDFs discussed in §D.5 and other multimodal media.

# E  QUALITATIVE ANALYSIS

We provide examples of the retrieved passages for a query from MMLU Pro in Table 28, GPQA in Table 29, and MATH in Table 30, respectively.

With ANN-only, the retrieved passages are relevant to the domain of the query but can sometimes be irrelevant to the query itself. Table 28 asks for the an incompatible application for hash tables, while the retrieved passage only mentions the mechanism of hash table without providing useful information mention for answering the question. In Table 29, the query asks for a calculation of ratio of numbers hydrogen atoms in the second excited state in the extreme temperature in the atmosphere of Sirius. The retrieved passage mentions the temperature in Sirius but are about wavelength, without mentioning hydrogen atoms at all. Table 30 shows another example where the query is about probability of Michael rolling three dice and getting at least two 1s. The retrieved passage with ANN only mentions Michael rolling dice, but it is about the probability distribution of the sum of two dice.

When using GRIT for Exact Search, the retrieved passages become more relevant to the queries themselves. For example, the retrieved passage in Table 29 includes a question and solution that is similar to the original query asking about the ratio of the number of hydrogen atoms in the first excited state. The one in Table 30 also mentions the joint probability of two dice throws.

LLM Reranking inconsistently provide more relevant passage than Exact Search. For instance, the retrieved passage in Table 28 specifically mention that hash table is not a good choice for range search. Passages that including such direct answer for the query is the most effective ones suggested by the oracle retrieved document also shown in the table. However, in Table 29, the retrieved passage using LLM Reranking only provide a relevant high level background of hydrogen atoms' excited state around Sirious in the extreme temperature without a providing a specific solution to the query.

**Comparing Retrieved Passages between COMPACTDS and Search Engine.** We additionally compare examples of retrieved passages between COMPACTDS-LLM and search engine (Aggregate→Rerank) for queries from MMLU Pro, GPQA, and Math in Tables 31,32, and 33, respectively. We also include a comparison between oracle-reranked retrieved passages from COMPACTDS and search engine for MMLU Pro in Table 34.

We first observe that search engine retrieval typically results in documents that differ significantly in format and nature from the text retrieved from COMPACTDS and are often atypical to the pre-training corpora that COMPACTDS is composed of. For example, Table 33 shows retrieved passages for a query from MATH which involves computing the volume of a sphere when given the volume of the circumscribing cylinder; while COMPACTDS yields a Stack Exchange post discussing how to derive the exact relationship between the volumes of a sphere and its circumscribing cylinder, the search engine yields a PDF worksheet[5] from a K-12 textbook which also derives the same formula alongside many other relationships. Table 34 shows retrieved passages from a query in MMLU Pro about deed validity. Both passages capture the necessary information regarding participation and intention in deed delivery. However, COMPACTDS's passage is from a related court case on the legitimacy of a property sale, while the search engine result comes from a guide about estate planning for Oklahoma farm and ranch families[6].

We also note that the lexical matching strength of search engine retrieval is inconsistently beneficial. Table 32 shows a question about the interpretation of the commutator of two gamma matrices. While the result from COMPACTDS is closely related to the given problem, but not completely sufficient to solve the problem, the search engine result exactly contains the commutator expression. Furthermore, it identifies how it contributes to the angular momentum of the Dirac field and derives the Lorentz transformations as well. On the other hand, Table 31 gives an example query that uses Euler's polyhedron formula. While COMPACTDS yields a result that exactly contains the required formula and how to use it, the search engine result is lexically noisy and contains numerous unrelated formulas for different 3D shapes. One possibility is that the short question and large number of answers could be distracting during the chunk CONTRIEVER-reranking stage. Nonetheless, the inclusion of such a document within the top-10 search engine results raises caution regarding search engine susceptibility to lexical distractions.

---

[5] https://static.bigideasmath.com/protected/content/pe/hs/sections/geo_pe_11_08.pdf

[6] https://extension.okstate.edu/fact-sheets/estate-planning-a-simplified-guide-for-oklahoma-farm-and-ranch-families.html

Table 25: Instruction for generating helpfulness score for LLM Reranking.

```
# Instruction
You are an expert evaluator. Your task is to evaluate how much
↪  the context can help solve the question and arrive at the
↪  correct answer.
We will provide you with the question and the context. You
↪  should first read the question carefully, and then evaluate
↪  the helpfulness of the context based on the scoring criteria
↪  provided below.

# Question
{full_text}

# Context
{retrieval_text}

# Scoring Criteria
Before outputting the score, provide a short reason for the
↪  decision, citing specific chunks of text from the context if
↪  applicable. Output the score in the range of 1~10, where 1
↪  means the response is extremely unhelpful and 10 means the
↪  response is extremely helpful.
Here are more detailed criteria for the scores:

- Score 1~2: The provided context is largely off-topic and
↪  provides minimal or no helpful information. Its content is
↪  very distant from the question at hand.
- Score 3~4: The provided context has a weak connection to the
↪  problem. While it may mention related concepts or offer minor
↪  insights, it does not contribute meaningfully to solving the
↪  question.
- Score 5~6: The provided context contains some relevant
↪  information, but it doesn't directly help in solving the
↪  question. It may provide background context or partial
↪  information that needs further clarification.
- Score 7~8: The provided context is highly relevant and
↪  addresses most aspects of the question. It provides clear
↪  and actionable information, though there may still be minor
↪  gaps or missing details.
- Score 9~10: The provided context is entirely relevant and
↪  offers thorough, accurate, and comprehensive information
↪  that directly solves the question. It covers all aspects
↪  necessary to fully address the question with precision.

Please output your reason and score as a JSON object.
```

Table 26: Instruction for generating sub-query breakdown.

```
{query}\n\nRewrite the above question as up to three unique
↪   search queries to use with a search engine to find helpful
↪   relevant information to solve the above problem. Only output
↪   the generated search queries as a json dict with key
↪   "search_queries" pointing to the list of generated search
↪   queries. Do not exceed three search queries.
```

Table 27: Instruction for identifying duplication.

```
Below is a question and answer pair and a retrieved
↪   passage.\nAnswer 'Yes' if: the passage contains the exact
↪   same question and answer, ignoring minor format
↪   difference.\nAnswer 'No' if: the passage does not contain
↪   either the exact same question or the answer, or the answer
↪   has to be summarized or inferred from the passage, or the
↪   answer has to be selected among other options.\nYou answer
↪   should only be 'Yes' or 'No'.\n\n<q&a>\n{question and
↪   answer}\n</q&a>\n\n<passage>\n{passage}\n</passage>\n\nYour
↪   answer:
```

Table 28: Examples of the top retrieved passage for MMLU Pro from COMPACTDS.

| | |
|---|---|
| Query | Question: Hash tables can contribute to an efficient average-case solution for all of the problems described below EXCEPT:
 A. Key comparison: Given two keys, determine if they are identical or different.
 B. Counting distinct values: Given a set of n keys, determine the number of distinct key values.
 C. Dynamic dictionary: Support the operations of insert, delete, and search in a dictionary.
 D. Range search: Given values a and b, find all the records whose key value is in the range a, b.
 E. Symbol table lookup: Given a program identifier, find its type and address.
 F. Frequency count: Given a set of n keys, determine how many times each key appears.
 G. Unique key identification: Given a key, find if it is unique in the set.
 H. Sequential search: Given a key value, find the first occurrence of the key in the set. |
| Answer | D |
| ANN only | ... Making use of the whole key gives a better chance of limiting the number of collisions. – try to spread the keys uniformly over the table. Avoid a function which deliberately biases keys to a certain portion of the table. The efficiency of the hashing technique depends mainly on the hashing function and the method of resolving collisions. The method described above is called the \textbf{linear} method and it gives good results provided that the table is no more than about 90\% full... |
| +Exact Search | ...compute the average number of comparisons needed to perform a successful search for a value in the table. 13. When a computer program is compiled, the compiler builds a symbol table for storing information about the identifiers used in the program. A scheme is needed to quickly decide whether a given identifier has already been stored in the table and, if not, to store the new identifier. A hash function is often used to locate a position in the table at which to store information about an item... |
| +LM Reranking | ...If you need to quickly find the maximum or minimum key, find keys in a given range, or implement any of the other operations in the ordered symbol–table API on page 366, then hashing is not appropriate, since these operations will all take linear time... |
| Oracle | ...The search efficiency of the range is very low. For example, to find the value between key001 and key002, you need to traverse all the keys to find the corresponding answer... |

Table 29: Examples of the top retrieved passage for GPQA from COM-
PACTDS.

| | |
|---|---|
| Query | Question: Sirius is the brightest star in the sky. The temperature of this star is around 10000 K. Consider Hydrogen atoms in the atmosphere of Sirius. What is the ratio of the number of hydrogen atoms in the second excited state of Hydrogen to those in ground state?
Choices:
 (A) 8.2 * 10**-8
 (B) 7.26 * 10^-6
 (C) 8.11 * 10^-7
 (D) 5.4 * 10**-9 |
| Answer | C |
| ANN only | ...The brightest star in the sky is Sirius, the Dog Star. It is actually a binary system of two stars, the smaller one (Sirius B) being a white dwarf. Spectral analysis of Sirius B indicates that its surface temperature is 24,000 K and that it radiates energy at a total rate of \(1.0 \times 10^{25}\) W. Assume that it behaves like an ideal blackbody. (a) What is the total radiated intensity of Sirius B? (b) What is the peak-intensity wavelength? Is this wavelength visible to humans? (c) What is the radius of Sirius B? Express your answer in kilometers and as a fraction of our sun's radius... |
| +Exact Search | ...The temperature of the surface of a certain star is 8000 K. Most hydrogen atoms at the surface of the star are in the electronic ground state. What is the approximate fraction of the hydrogen atoms that are in the first excited state (and therefore could emit a photon)? The energy of the first excited state above the ground state is (-13.6/22 eV) - (-13.6 eV) = 10.2 eV = 1.632e-18 J. 2. Relevant equations Kb = 1.38e-23 J/K 1/T = E/Kb 3. The attempt at a solution I don't really know how to do this problem. My guess was to divide the energy in joules by the temperature and then divide by the boltzmann constant to make it unitless. (1.632e-18/8000)/(1.38e-23)=14.8 2. Apr 28, 2010 ### nickjer Hmm.... You are right so far, but you might want to use the Boltzmann factor for this problem as well. 3. Apr 28, 2010 ### burg25 What do you mean? Could you clarify? 4. Apr 28, 2010 ### burg25 Wait I think I understand. Are you saying take e raised to this value? 5. Apr 28, 2010 ### burg25 Yeah I got it. Thanks 6. Apr 28, 2010 ### nickjer Yes, using the Boltzmann factor you get: $$\frac{P(E_1)}{P(E_0)} = e^{-\frac{E_1-E_0}{kT}}$$ where $P(E)$ is the probability the electron is in the state with energy $E$. |
| +LM Reranking | ...Can we conclude that the atmosphere of Sirius is made up largely of hydrogen while the solar atmosphere consists largely of metal atoms? Since the early 1920s astronomers have understood that the most conspicuous differences between stellar spectra arise from differences in the temperature of the atmospheric layers where the spectral lines are formed, rather than from differences in the relative abundances of the chemical elements. To absorb light at the frequency of one of the Balmer lines, a hydrogen atom must be in its first excited state. The fraction of hydrogen atoms in this state depends on the temperature (and, much more weakly, on the pressure). At the temperature of the solar photosphere (the visible layer of the Sun's atmosphere), nearly all of the hydrogen atoms are in the ground state, where they can absorb lines of the ultraviolet Lyman series. The spectrum of Sirius is formed at a temperature of around 10,000 K. A much larger (though still numerically small) fraction |

Table 30: Examples of the top retrieved passage for MATH from COM-PACTDS.

| | |
|---|---|
| Query | ```
Problem:
If Michael rolls three fair dice, what is the probability that he
will roll at least two 1's? Express your answer as a common
fraction.

Solution:
``` |
| Answer | `\\frac{2}{27}` |
| ANN only | ```
brother Michael want to borrow their father's car on a Friday
night. To determine who gets to use the car, Carmen wants her
father to roll a pair of fair dice. If the sum of the two dice is
2, 3, 11, or 12, Carmen gets to use the car. If the sum of the two
dice is 4 or 10, then Michael can use the car. If the sum is any
other number, then the dice will be rolled again. Michael thinks
that this is not a fair way to decide. Is he correct? Explain. No.
Michael is wrong. The probability of rolling a sum of a 2, 3, 11,
or 12 is $$\frac{1}{21}$$ + $$\frac{1}{21}$$ + $$\frac{1}{21}$$ +
$$\frac{1}{21}$$ = $$\frac{4}{21}$$. The probability of rolling a
sum of a 4 or 10 is $$\frac{2}{21}$$ + $$\frac{2}{21}$$ =
$$\frac{4}{21}$$. Because both Carmen and Michael are equally
likely to get the car, this would yield a fair decision. Question
2. Due to a technology glitch, an airline has overbooked the
number of passengers in economy class on a flight from New York
City to Los Angeles. Currently, there are 150 passengers who have
economy class tickets, but there are only 141 seats on the plane.
There are two seats available in first class and one seat
available in business class. a. Explain how the ticket agent could
use a random number generator to make a fair decision in moving
some passengers to either the first - or business - class sections
of the plane and to
``` |
| +Exact Search | ```
Problem: Nathan will roll two six-sided dice. What is the
probability that he will roll a number less than three on the
first die and a number greater than three on the second die?
Express your answer as a common fraction. Solution: For the first
die to be less than three, it must be a 1 or a 2, which occurs
with probability $\frac{1}{3}$. For the second die to be greater
than 3, it must be a 4 or a 5 or a 6, which occurs with
probability $\frac{1}{2}$. The probability of both of these events
occuring, as they are independent, is $\frac{1}{3} \cdot
\frac{1}{2} = \boxed{\frac{1}{6}}$.
``` |
| +LM Reranking | ```
Q: If you roll three dice, what is the probability of getting at
least two number are same? If you roll three dice,What is the
probability of getting at least two numbers the same? A: Hint:
compute the chance they are all different, then subtract from $1$.
``` |

Table 31: Example of top retrieved results from COMPACTDS and search engine retrieval for a query from MMLU Pro (Computer Science).

| | |
|---|---|
| Query | Question: Suppose a convex 3d-object has 15 vertices and 39 edges. How many faces does it have?
A. 25
B. 26
C. 20
D. 31
E. 22
F. 28
G. 32
H. 30
I. 24
J. 27
Answer: |
| Answer | B |
| COMPACTDS | ... Each triangular prism has a base area of \\( \\frac{1}{2}(8)(5.5) \\) or 22 cm\u00b2 and a height of 10 cm. 41a. Sample answer: 49. semicircle; 180 51. major arc; 270 53. 73 mm, 180.5 mm\u00b2 41b. Greater than; a square with a side length of 6 m has an area of 36 m\u00b2. A circle with a diameter of 6 m has an area of 9\u03c0 m\u00b2. Since the heights are the same, the volume of the square prism is greater. 41c. Multiplying the radius by x; since the volume is represented by \u03c0r\u00b2h, multiplying the height by x makes the volume x times greater. Multiplying the radius by x makes the volume x\u00b2 times greater. 43a. base 3 in. by 5 in., height 4\u03c0 in. 43b. base 5 in. per side, height 12\u03c0 in. 43c. base with legs measuring 3 in. and 4 in., height 10\u03c0 in. 45. Sample answer: 47. Both formulas involve multiplying the area of the base by the height. The base of a prism is a polygon, so the expression representing the area varies, depending on the type of polygon it is. The base of a cylinder is a circle, so its area is \u03c0r\u00b2. 49. F 51. C 53. 126 cm\u00b2; 175 cm\u00b2 55. 205 in\u00b2 57. 11.4 cm 59. 9.3 in. 61. 378 m\u00b2 Lesson 12-5 1. 75 in\u00b3 3. 62.4 m\u00b3 5. 51.3 in\u00b3 7. 28.1 mm\u00b3 9. 513.333 ft\u00b3 11. 105.8 mm\u00b3 13. 233.8 cm\u00b3 15. 35.6 cm\u00b3 17. 235.6 in\u00b3 19. 1473.1 cm\u00b3 21. 1072.3 in\u00b3 23. 234.6 cm\u00b3 25. 32.2 ft\u00b3 27. 3190.6 m\u00b3 29. about 13,333 BTUs 31a. The volume is doubled. 31b. The volume is multiplied by 2\u00b2 or 4. 31c. The volume is multiplied by 2\u00b3 or 8. 33. 14 in. 35a. Sample answer: 35b .... |
| Search Engine | Q: A formula to describe the relation of faces, edges and vertices in three-dimensional convex bodies Is there a formular, and if yes, what is it, to describe the relation of faces, edges and vertices in three-dimensional convex bodies. for regular shapes: A tetrahedron has 4 faces, 6 edges and 4 vertices Cube: 6 faces, 12 edges, 8 vertices Octahedron: 8 faces, 12 edges, 6 vertices Pentagonal dodecahedron: 12 faces, 30 edges, 20 vertices What about a n-faced polyhedron? n faces, but how many edges and vertices? Is there a formula to calculate the number of vertices and edges, given a specific number of faces? Or a range of possible numbers of vertices and edges? Add-on: What happens under the assumption of irregular shapes with that formula? A: Yes, there is such a formula. It is called Euler's characteristic formula, and it states that if $V$ is the number of vertices, $E$ the number of edges, and $F$ the number of faces of a polyhedron, then $$V-E+F=2$$ For example, the cube has $8$ vertices, $6$ faces and $12$ edges, and $8-12+6=2$. The octahedron has $6$ vertices, $8$ faces and $12$ edges, and again $6-12+8=2$. A: Euler's formula even allows dimensional regression. Let $F=\\{f_0, f_1, f_2, \u2026\\}$ be the facet vector of your polytope $P$, i.e. $P$ has $f_0$ $0$-facets (vertices), $f_1$ $1$-facets (edges), $f_2$ $2$-facets (faces), etc. Then you have $$\\sum_{k=0}^{d-1} (-1)^k f_k = 1-(-1)^d$$ That formula also holds for non-regular polytopes, provided you do not encounter holes and other odd stuff. Esp. it is valid |

Table 32: Example of top retrieved results from COMPACTDS and search engine retrieval for a query from GPQA (Physics)

| | |
|---|---|
| Query | Question: Which of the following statements is a correct physical interpretation of the commutator of two gamma matrices, i/2 [gamma^mu, gamma^nu]?

1. It gives a contribution to the angular momentum of the Dirac field.
2. It gives a contribution to the four-momentum of the Dirac field.
3. It generates all Poincar\u00e9 transformations of the Dirac field.
4. It generates all Lorentz transformations of the Dirac field.
Choices:
(A) 1 and 4
(B) 2 and 4
(C) 2 and 3
(D) 1 and 3 |
| Answer | A |
| COMPACTDS | P_\\mu = i \\frac{\\partial}{\\partial x^\\mu}. \\tag{3.51} \\]
From the relations derived it follows that \\(hM_k\\) is the orbital angular momentum projection operator whereas the translation operator, multiplied by \\(h\\), represents the four-dimensional energy-momentum vector \\(p_\\mu\\). One can be easily convinced of the fact, that the operators \\(M_{\\mu\\nu}\\) and \\(P_\\mu\\) satisfy the commutation relations (3.46)\u2013(3.48). Relations (3.47) and (3.48) demonstrate, that the generators \\(M\\) and \\(P\\) commute with the Hamiltonian \\(P_0\\), what allows us to classify physical states according to eigenvalues of these generators. The eigenvalues of the Lorentz boosts generators \\(N\\) cannot be used for this purpose because they do not commute with \\(P_0\\). In case of the finite-dimensional Poincare transformations, the law of fields transformation has the form: \\[ \\psi'_i(x) = B_i^j(\\Lambda) \\psi_j[(\\Lambda^{-1}x - a)], \\] where the set of matrices \\(B(\\Lambda)\\) forms representations of the Poincare group. Once again to classify all irreducible unitary representations of the group one must find all the representations of the permutation relations (3.46)\u2013(3.48) in the form of Hermitian operators. To achieve this goal we fix Casimir operators. Let us define the so-called Pauli-Lubanski pseudovector: \\[ W_\\sigma = \\frac{1}{2} \\varepsilon_{\\sigma\\mu\\nu\\lambda} M^{\\mu\\nu} p^\\lambda. \\tag{3.52} \\] In Eq. (3.52) \\(M^{\\mu\\nu}\\) is a generator of representation of the proper Lorentz group, and consequently, it is already a sum of both orbital and spin moments of a wave field. In the three-dimensional vector notations \\(W_\\sigma\\) has the form: \\[ W^0 = p \\cdot M, \\quad W = p_0 M - [p \\times N]. \\tag{3.53} \\] It is obvious that the four-dimensional pseudovector \\(W_\\sigma\\) |
| Search Engine | Pauli matrices with two spacetime indices \u2022 #1 John Corn 2 0 \"Pauli matrices with two spacetime indices\" Hi all. This is my first post so forgive me if my latex doesn't show up correctly. I am familiar with defining a zeroth Pauli matrix as the 2x2 identity matrix to construct a four-vector of 2x2 matrices, $\\sigma^\\mu$. I'm trying to read a paper which uses the notation $\\sigma^{\\mu \\nu}$. This is between a 4-spinor and a gamma matrix. Can someone please enlighten me about what this notation means? Thanks so much. Physics news on Phys.org \u2022 #2 I vaguely remember it to be the (anti-?) commutator of two gamma matrices. \u2022 #3 Thanks for the quick response Dr. Du. The anticommutator of gamma matrices is just $2 \\eta^{\\mu \\nu} I_{4 \\times 4}$, which hardly calls for new notation. One usually doesn't discuss commutators in relation to Clifford algebra, but I can't rule that out. \u2022 #4 As far as I remember [tex] \\Sigma^{\\mu\\nu} := \\frac{i}{2}\\left[\\gamma^{\\mu},\\gamma^{\\nu}\\right]_{-} [/tex] It has to do with the spin operator for the quantized massive Dirac field. \u2022 #5 The Sigma matrices are usually used during the derivation of the Lorentz covariance and transformation properties of the Dirac equation. Later it is usually shown how to represent the Sigma matrices using thre gamma matrices. So strictly speaking you don't need them (or you only need them in an intermediate step) Back Top |

Table 33: Example of top retrieved results from COMPACTDS and search engine retrieval for a query from MATH, truncated for viewing.

| | |
|---|---|
| Query | Problem:
The volume of a cylinder is 60 cubic centimeters. What is the number of cubic centimeters in the volume of the sphere it circumscribes?

Solution: |
| Answer | 40 |
| COMPACTDS | Q: A cylinder is circumscribed about a sphere. If their volumes are denoted by $C$ and $S$, find $C$ as a function of $S$ Here is the problem. A cylinder is circumscribed about a sphere. If their volumes are denoted by $C$ and $S$, find $C$ as a function of $S$ My (Amended) Attempt: [Based on the correction suggested by herbSteinberg] Let $r$ be the radius of the sphere. Then the height of the cylinder is $2r$, and the radius of the base is $r$. So the volume $C$ of the cylinder is given by $$ C = \pi r^2 (2r) = 2 \pi r^3. \tag{1} $$ And, the volume $S$ of the sphere is given by $$ S = \frac43 \pi r^3. \tag{2} $$ From (2), we obtain $$ r^3 = \frac{3}{4 \pi} S = \frac{ 3S }{4 \pi}, $$ and hence $$ r = \sqrt[3]{ \frac{3S}{4 \pi} }. \tag{3} $$ Finally, putting the value of $r$ from (3) into (1), we get $$ C = 2 \pi \left( \sqrt[3]{ \frac{3S}{4 \pi} } \right)^3 = 2 \pi \left( \frac{3S}{4 \pi} \right) = \frac32 S. $$ Is my solution correct in each and every detail? Or, are there any errors of approach or answer? A: As noted in the comments, the radius of the cylinder is just $r$. Otherwise your work appears to be correct, but you've made things harder on yourself than necessary. Note that $r^3$ appears in both formulas, so once you have solved for $r^3 = \frac{3S}{4\pi}$, you can immediately use this expression |
| Search Engine | Volumes of Spheres Essential Question How can you find the volume of a sphere? EXPLORATION 1 Finding the Volume of a Sphere Work with a partner. A cylinder is circumscribed about a sphere, as shown. Write a formula for the volume $V$ of the cylinder in terms of the radius $r$. $$V = \text{Volume of cylinder}$$ When half of the sphere (a hemisphere) is filled with sand and poured into the cylinder, it takes three hemispheres to fill the cylinder. Use this information to write a formula for the volume $V$ of a sphere in terms of the radius $r$. $$V = \text{Volume of a sphere}$$ Use the Internet or some other resource to confirm that the formula you wrote for the volume of a sphere is correct. EXPLORATION 2 Finding the Volume of a Sphere Another Way Work with a partner. The figure shows a hemisphere, and a cylinder with a cone removed. A plane parallel to their bases intersects the solids $z$ units above their bases. Using the AA Similarity Theorem, the radius of the cross section of the cone at height $z$ is $z$. The area of the cross section formed by the plane is $\pi(r^2 - z^2)$ for both solids. Because the solids have the same height and the same cross-sectional area at every level, they have the same volume by Cavalieri\u2019s Principle. a. Write the relationship between the volume of the hemisphere, the volume of the cylinder, and the volume of the cone. b. Use the relationship in part (a) to derive the formula for the volume of a sphere with a radius $r$. Communicate Your Answer 3. How can you find the |

Table 34: Example of top retrieved results from COMPACTDS and search engine retrieval for query in MMLU Pro (Law), with oracle reranking applied. Documents are truncated for viewing.

| | |
|---|---|
| Query | Question: Two brothers owned a parcel of real estate as joint tenants. Both brothers signed a deed as grantors conveying the land to the buyer. The first brother handed the deed to the second brother with instructions to take the deed to their lawyer for approval prior to delivery. Brother two, without his brother's permission, took the deed directly to the buyer and collected the sale price. Is this a good deed as to partner one?
 A. Yes, the deed was signed by both partners, which proved their intent to sell to buyer, and it was delivered at the time of signing by the fact of affixing their signatures to the document.
 B. Yes, the deed is valid as the buyer was not aware of the internal agreement between the two brothers.
 C. No, the deed is invalid as partner one did not give explicit permission for partner two to deliver the deed.
 D. Yes, the deed is valid as partner two had the authority to finalize the deal.
 E. No, the deed is invalid because partner two collected the sale price without partner one's consent.
 F. Yes, the deed is valid as both brothers had signed it, signifying their agreement to the sale.
 G. Yes, the transfer is valid from both partners because partner two was partner one's apparent agent for purposes of delivering the deed.
 H. No, the deed was invalid as to both grantors because partner two stepped outside his scope of authority.
 I. No, the deed cannot bind partner one because he did not participate in the deed delivery to the buyer and did not intend to deliver the deed up to the grantee at that time.
 J. No, the deed is not valid as the lawyer did not approve it before delivery.
 Answer: |
| Answer | I |
| COMPACTDS | ... The absence of a formal deed of conveyance is a strong indication that the parties did not intend immediate transfer of ownership. Fourth, petitioner retained possession of the certificate of title of the lot. This is an additional indication that the agreement did not transfer to private respondents, either by actual or constructive delivery, ownership of the property. Finally, respondent Juanito admitted during trial that they have not finalized the sale in 1972 because there were minor owners such that when they constructed their house thereon, they sought the permission of petitioner. Now, the next question to be resolved is whether the suspensive condition, i.e., judicial approval of the sale of the minor owners' shares, upon which the obligation of the sellers to execute a deed of sale depends, is fulfilled. Article 1186. The condition shall be deemed fulfilled when the obligor voluntarily prevents its fulfillment. This provision refers to the constructive fulfillment of a suspensive condition, whose application calls for two requisites, namely: (a) the intent of the obligor to prevent the fulfillment of the condition, and (b) the actual prevention of the fulfillment. Mere intention of the debtor to prevent the happening of the condition, or to place ineffective obstacles to its compliance, without actually preventing the fulfillment, is insufficient.... |
| Search Engine | ... Deeds are recorded in the County Clerk\u2019s office of the county in which the land is located. \u2022 Importance of Delivery. Sometimes a grantor will sign a deed covering land which he or she wants to keep until death expecting the deed to be the method by which title to the land is transferred. Instead of delivering the deed directly to the grantee, the owner places the deed in a safety deposit box or among private papers. Undelivered deeds like these are commonly called \u201cdresser drawer deeds.\u201d Such deeds are invalid because they were not delivered to the grantee or his or her agent during the lifetime of the grantor. Delivery is necessary to accomplish a conveyance by deed. Unless the deed is drawn and executed in the form of a will (which would be extraordinary and unusual), it could not become legally effective without delivery of the deed to the grantee or his agent... The key to effective delivery of a deed is that the grantor must presently transfer legal control over the deed to the grantee or an independent third party. If the grantor retains a right to change his mind, then control has not been presently transferred. Because of the potential legal problems that may arise in using this type of device in estate planning, it is especially important to obtain legal advice... |

