# OpenReview forum: "Frustratingly Simple Retrieval Improves Challenging, Reasoning-Intensive Benchmarks"
_ICLR.cc/2026/Conference — ICLR 2026 Poster_

### Official Review · Reviewer_41Bx · 2025-10-31

**Soundness:** 1
**Presentation:** 2
**Contribution:** 2
**Rating:** 4
**Confidence:** 4

**Summary:**

This paper introduces CompactDS for RAG in LLMs, a web-scale datastore built from filtered web crawl CommonCrawl plus specific math, academic, and educational texts to better cover content relevant to the evaluation tasks. To efficiently access the datastore, the paper proposes a two-stage retrieval mechanism: on-memory approximate nearest neighbor search using an build index followed by on-disk exact search. The aggressive content filtering enables single-node low-latency RAG use because the full datastore can fit <500GM RAM device. The paper evaluates a simple RAG pipeline on CompactDS on reasoning benchmarks like MATH and GPQA and language understanding (MMLU), and releases the datastore for public use.

**Strengths:**

1. The paper is generally well-written and clear.
2. Open-sourcing and publicly releasing the datastore will be a plus for the research community when the industry datastores are proprietary and behind paywalls.
3. MassiveDS and other non-filtered datastores can be impractical due to large memory footprint; a compact datastore like CompactDS will help the research community.
4. The paper compares various approaches for retrieval-based generation, not just embedding-based RAG but also agentic systems based on classical search engines. This grounds the paper's contribution in several popular approaches.

**Weaknesses:**

1. While the paper explicitly describes the data sources in Sec 3.1, the justification for each filtering step is unclear. For example, in line 134... why take the union of two datasets then filter using a classifier with a certain threshold? Is the target to reduce to a maximum number of words in the corpora, or to maximize some validation split in the evaluation setup? This problem persists in the whole of Sec 3.1, which is a key contribution of the paper. Without this justification, design choices for data filtering seem arbitrary and do not help build intuition on what makes a datastore "good" or "bad".
2. The summary in Sec 3.2 is not fully clear, but could improve significantly with a visual diagram spanning the top of this page. Specifically, it's unclear to me which K disk-stored embeddings are retrieved in the 2nd stage of the 2-stage process.
3. It is difficult to be convinced that the 2-stage process is significantly better than the 1-stage (ANN only) process, comparing rows CompactDS-ANN only and CompactDS in Table 1. MMLU doesn't really improve and GPQA varies with different parts. There is improvement in MMLU Pro, AGI Eval, and MATH though. It would be useful if the authors discuss and interpret these results further.
4. Since the paper argues that MassiveDS is too massive, which makes it impractical, it makes sense to evaluate it in Table 1 and report all benchmarks (not just MMLU in Table 2).
5. Significant drops in performance on GPQA and MATH after decontamination (Sec A.1) suggest that CoT is more important for reasoning than retrieving passages. This counters the paper's argument that retrieval could be important for math reasoning tasks.

**Questions:**

1. Line 68: Why does removing weakest contributing sources degrade performance? One would expect that the most similar documents would still be retrieved, and so the performance would be unaffected. Why is "long tail diversity" important (line 284)?
2. Line 116: While this work does not focus on agentic RAG, do you think it can still work with RAG in an interactive setting? Say when the agent requests a tool call/search string but we retrieve non-exact document hits (e.g., in cases where the search string doesn't exist in the document corpora verbatim)?
3. Line 192: What is the time and space overhead at training time, for indexing with IVFPQ and exact embeddings?
4. Line 197: Need some clarity in understanding the 2-stage process. After q is encoded as E_exact(q), which K disk-stored embeddings are used? Are these the K passages retrieved in the 1st stage of IVFPQ? If that is so, wouldn't the lossy IVFPQ retrieval hurt the second stage? If that's not the case, then finding K passages with exact search in the full index would be extremely time-intensive.
5. Table 1: CompactDS vs No Retrieval on Bio is worst (39.7 vs 47.4) but the relative gain is still positive? This seems incorrectly computed, or a typo.
6. Line 250: good to point to Table 7 with the search-o1 results.
7. Table 4: GPQA scores are consistently worse for CompactDS compared to NoRetrieval for all LLMs except Llama3.1 8B. Is there any insight into this? To me, arguing that CoT is extremely competent at 70B scale to improve NoRetrieval is not a great argument.
8. Line 303 and Table 3: What is the oracle reranking? This needs explanation here.
9. Line 410 and Table 6: If the PDFs cannot fit in context at all, how do we get evaluation results for the rows without LM Reranking?

### Typos and editorial suggestions
1. Line 171: spacing in argTopk...
2. Table 1: it would make sense to only present No Retrieval, CompactDS rows, and any other baselines in this table. Then another table for ablation on single-source datastores.
3. Line 248: demends -> demands?
4. Line 256: point to the appendix with other LLM results.
5. Fig 5: the font is extremely small even on my large monitor. I cannot read the math expressions at all. I can barely read the plots elsewhere, but this plot is really hard even after zooming in.

---

> ### Author Response · Authors · 2025-11-21
> **Part 1/3 of the Response to Reviewer 41Bx**
>
> We thank the reviewer for their helpful feedback, and for supporting our work!
>
> **W1. Dataset Filtering:**
> We will revise the paper to provide better justification of our data sources and filtering pipeline. A summary is provided below. Our pipeline largely follows best practices in modern LLM data curation, and draws heavily from recent high-quality open-source training efforts.
>
> - Web data: We included the web data as it has been the de-facto source for broad coverage and diversity. We use both C4 and DCLM-Baseline, because they’re known to be the highest-quality Common Crawl based datasets available. However, both rely on relatively simple heuristic and fastText-based filtering. For this reason, we apply an additional layer of filtering based on FineWeb-Edu’s approach, which uses a more sophisticated classifier based on Llama-3.3-70B-Instruct’s quality annotations. We do this instead of taking FineWeb-Edu data directly because FineWeb-Edu’s own text is known to be slightly lower quality due to a less robust text extraction pipeline. A similar multi-stage pipeline has been adopted in prior work, such as OLMo 2’s mid-training data curation [1].
>
> - Wikipedia and Books: We include them because they have long been recognized as high-quality corpora for pretraining. For instance, BERT is trained on Wikipedia and Books.
>
> - Expert Data: We include expert domains such as math, academic papers, and code, as strong reasoning capabilities often require exposure to these domains. For each domain, we use the most up-to-date, carefully curated public datasets.
>
> - Q&A forums: We included Q&A forums as these sources are widely regarded as high-quality for training and often contain well-structured problem-solution pairs.
>
> **W2. Visual Diagram for Retrieval Pipeline**:
> [Here](https://anonymous.4open.science/api/repo/ICLR-image-4722/file/Blank%20diagram.png) we provide a figure of our two stage retrieval pipeline (anonymized). At the second stage, for top K most relevant documents retrieved from IVFPQ index, we find their embeddings saved on disk offline (GritLM-7B), and select the top k passages among these K passages that have the highest similarity with the query. We will add this figure to the next revision.
>
> **W3. Comparison between Stage 1 and 2**: It is relatively easier to retrieve helpful passages for MMLU, which makes the impact of a better retrieval less obvious. For GPQA, the variance of the performance has been high for our experiments. This is why we report the average over the benchmarks to get a more holistic view of improvements over the different pipelines. As the summarized table below suggests, the average performance across five benchmarks clearly raises from applying Exact Search (stage 2).
> | | ANN-Only| ANN+Exact Search |
> |---|---|---|
> | $k=3$ | 52.2   | 54.1 |
> | $k=10$ | 53.8   | 55.1  |
>
> **W4. Reproducing MassiveDS**: While we agree with the reviewer’s suggestion, unfortunately, due to the size of MassiveDS, we cannot run MassiveDS locally to evaluate over the whole benchmark suite. The MassiveDS authors did share their cached retrieval top-k=1000 results over MMLU from which we derived our reported results, but they too had taken down their instance of MassiveDS due to its resource consumption, and were unable to run search queries for our other benchmarks. This further highlights the need for academically accessible datastores to study RAG over pretraining-corpora-breadth datastores.
>
> **W5. Performance Drops from Decontamination**:
> First, we clarify that the figure in A.1 actually shows that performance on MATH and GPQA go up after decontamination. We also clarify that our demonstration of improvements through retrieval is after enabling CoT, indicating that, while CoT is important, adding retrieval on top of it yields further gains. Our goal is not to claim that retrieval is more or less important than CoT, but rather to emphasize that these two are complementary, and retrieval provides benefits even on top of CoT.

---

> > ### Author Response · Authors · 2025-11-21
> > **Part 2/3 of the Response to Reviewer 41Bx**
> >
> > **Q1. Why is Long-tail Diversity Needed**: While the weakest contributing sources don’t demonstrate the best aggregate performance over all the benchmarks, many sources still contain documents that are helpful for specific questions. Once these weaker sources are removed, those documents become unavailable and may not be replaceable by documents from the other sources, hence the drop in performance. This is largely because the sources we choose are diverse from each other, each contributing meaningful long-tail information (which is why we use the term long-tail diversity).
> >
> > **Q2. CompactDS’s potential in an Interactive Setting**: Because we embed all documents using embedding language models, even when the exact search string does not exist in the datastore, the retriever should be able to find the most semantically similar documents to the search string. While there might exist scenarios in an agentic setting where a precise response is needed (mathematically calculation) and RAG would not be the best choice, based on our experiment in Section 5, we believe that CompactDS serves a strong alternative candidate of web retrieval whenever the agent calls for information retrieval.
> >
> > **Q3. Time and Space Overhead at Training Time**: In our experiments, training the ANN index with samples of the data with an H100 GPU took approximately 3 hours, while constructing the index took over 50 hours. As stated in the paper, the resulting index takes 456 GB. For Exact Search, the time overhead lies in embedding the passages, which takes 2k retrieved passages / min.
> >
> > **Q4. Clarity on Two-stage Process**: The reviewer’s understanding of the two-stage process is correct. At the second phase where q is encoded as E_exact(q), the exact inner product search is conducted over the K passages (the exact embeddings stored on disk) retrieved during the IVFPQ first stage. It is entirely true that the first stage is lossy and could miss capturing relevant documents in the resulting pool of documents after the first-stage. This is a tradeoff between performance and efficiency. Using ANN with IVFPQ allows us to store the whole index in memory when conducting the first-stage retrieval at the cost of potentially losing retrieval accuracy. The second stage of retrieval is intended to recuperate retrieval accuracy by performing exact search over the smaller intermediary pool of documents. Section C.3 and Tables 12,13 demonstrate how our second stage retrieval recovers accuracy compared to one-stage ANN pipeline, and also demonstrates that using a weaker embedder (Contriever) for the first stage ANN retrieval and then using a stronger embedder (GRIT) for the second stage outperforms a single stage pipeline using just a strong embedder (GRIT), all while being more time and memory efficient. We also note that the reviewer’s observation that doing a single-stage exact search over the full index is exactly why MassiveDS was so computationally expensive at search time, even when search was run over distributed nodes.

---

> ### Author Response · Authors · 2025-11-21
> **Part 3/3 of the Response to Reviewer 41Bx**
>
> **Q7. Why do GPQA scores Change Inconsistently Across Models**: We give two complementary hypotheses for why change in GPQA performance is inconsistent for different models. Firstly, we acknowledge that GPQA is intended to be “Google-proof”, in that even a human evaluator with access to a search engine would have difficulty finding helpful documents; thus, it is likely that while retrieval can give helpful documents, it can also return many documents that are distracting and unhelpful. For weaker models, helpful documents can allow them to answer problems correctly that they originally couldn’t, while distracting documents end up not hurting overall performance (since the original model was already getting the problem wrong). On the other hand, for stronger models like Llama 3 70B which get more problems correct on their own, we hypothesize the increased sensitivity to distracting documents can result in the model getting problems wrong that it was originally getting correct. We believe that future work can help close this gap with higher quality retrievers and more discerning models.
> Furthermore, we suspect that long reasoning training undergone by models like Qwen3 8B make them less capable at using in-context documents (since reasoning tuning doesn’t involve the use of in-context documents).
>
> **Q8. Oracle Reranking Definition**: We agree with the reviewer’s suggestion and will make sure to clarify the definition of oracle reranking within the main paper in the next revision. Currently, the formal definition for oracle reranking can be found in C.6. We add the definition below as well: Given a test query q, the ground truth answer a, and $K$ candidate passages from COMPACTDS-ANN, the oracle passages are
>
> $p_1, p_2,\cdots, p_{k_\text{rerank}} = \mathrm{argTop}{k_\text{rerank}}{1 \leq j \leq K} (P_\text{LM}(a|p_j,q) - P_\text{LM}(a|q)) .$
>
> where $P_\text{LM}(a)$ represents the probability that the LM assign to the gold answer $a$. These oracle passages then replace the unranked passages in the generation stage.
>
> **Q9. How are Long Documents like PDFs handled in Search Engine Retrieval**: We clarify how PDFs and other long documents from web retrieval are parsed for in-context use. All documents are segmented into chunks of up to $c$ words. Then, all the chunks across the returned documents from web search are aggregated and embedded with CONTRIEVER. The query is also embedded with CONTRIEVER, and the aggregated chunks are reranked according to similarity with the query embedding. We then take the top $k$ chunks to feed in context. In our experiments we use $c = 512$ and $k = 3$. This is currently detailed in lines 411-414, but we will make sure this process is explained clearer in the next revision.
>
> **Typos and Editorial Suggestions (including Q5 and Q6)**: We appreciate the reviewer for carefully reading our paper and pointing them out. We will revise them in future revisions.

---

### Official Review · Reviewer_Uhfh · 2025-11-01

**Soundness:** 3
**Presentation:** 4
**Contribution:** 3
**Rating:** 8
**Confidence:** 4

**Summary:**

This paper revisits the assumption that retrieval-augmented generation (RAG) is ineffective on reasoning-intensive benchmarks. The authors build a large and diverse datastore called COMPACTDS, combining academic, mathematical, and educational sources, and evaluate a simple retrieval-augmented generation setup without complex pipelines. To make it deployable on a single node, COMPACTDS integrates in-memory approximate nearest-neighbor (IVFPQ) retrieval with on-disk exact search to balance recall and latency. The proposed approach consistently improves accuracy across reasoning benchmarks and model scales, showing that even a minimal retrieval system can enhance reasoning performance when supported by a sufficiently rich and diverse datastore. COMPACTDS also matches or outperforms commercial search engines (e.g., Google Search) and complex agentic RAG systems such as Search-o1.

**Strengths:**

The paper is clearly written and presents strong empirical evidence that retrieval remains useful for reasoning tasks when the datastore is comprehensive and high quality. The motivation is well grounded in an important and practical question. The retrieval design is simple yet effective, showing that careful data construction and indexing choices can yield significant improvements over prior baselines. Experiments are systematic, covering multiple model families and evaluation settings, and the results convincingly support the paper’s main claim. The analysis provides useful insight into how retrieval complements reasoning models. Releasing COMPACTDS as an open, reproducible resource would represent a meaningful contribution to the community.

**Weaknesses:**

The main limitation of the paper lies in its limited conceptual novelty. The contribution is primarily an engineering improvement through large-scale data construction rather than a new retrieval or reasoning algorithm. Although the results are strong, it remains unclear how well COMPACTDS generalizes beyond the specific reasoning benchmarks used in evaluation. Since the datastore composition is closely aligned with those benchmarks, the observed gains might partly reflect domain overlap rather than general reasoning improvement. It would be helpful to understand how the system performs on more general or non-reasoning tasks, and how flexible the datastore is when retrieval data need to be updated frequently, which is an important advantage of RAG systems in practical use.

**Questions:**

- How well does COMPACTDS generalize to new reasoning datasets or more general tasks outside the benchmarks used in this paper?
- How sensitive is the final performance to the strength and relative similarity between $E_{Approx}$ and $E_{Exact}$? Does the consistency of their retrieved results significantly influence performance?
- Is there a quantitative analysis of retrieval quality (e.g., precision and recall) to verify that improvements stem from better retrieval?
- Have the authors examined whether retrieval mainly aids factual grounding or intermediate reasoning steps?
- Did the authors attempt a fine-tuned retriever trained directly on reasoning tasks, and how would it compare to COMPACTDS?

---

> ### Author Response · Authors · 2025-11-20
> **Part 1/2 of the Response to Reviewer Uhfh**
>
> We thank the reviewer for their helpful feedback, and for supporting our work!
>
> **W1. Limited Conceptual Novelty**: We’d like to clarify that our work introduces novel combinations of techniques resulting in empirical findings that challenge existing assumptions in RAG research. Specifically:
> - To the best of our knowledge, previous work in data curation and processing (e.g., DCLM, FineWeb) has focused primarily on pre-training, and we are the first to apply them for retrieval.
> - Similarly, although ANN and Exact Search have been used independently, we are unaware of any work that shows combining them is essential for a good accuracy-efficiency tradeoff.
> - Our carefully ablated design of the datastore and RAG pipeline yields empirical results that challenge prevailing beliefs – that simple RAG is inadequate for challenging reasoning benchmarks.
> - Furthermore, we open-source (and host a version of) a web-scale datastore that is accessible, reproducible, and purpose-built for RAG research.
>
> The reviewer is correct that our work did not propose a new algorithm. However, We believe the novel combination of prior methods, along with new empirical findings, aligns well with the goals of the ICLR Call for Papers, which includes “datasets” and “infrastructure”. We are confident that our work will play an important role in making retrieval-based systems more accessible and practical.
>
> **W1+Q1. Domain Generalizability of CompactDS**: CompactDS is a web-scale datastore that covers a wide range of domains, a result of how we selected our data sources and because a large portion of it is high-quality web-crawled data, which we believe indicates that is more general than simply being aligned to our current benchmarks. In terms of evaluation, MMLU and MMLU Pro, with their wide range of topics, also cover more general and non-reasoning tasks in domains like politics and law. Both the performance improvement brought by CompactDS on these benchmarks and the composition of CompactDS itself are strong evidence to show CompactDS’s generalizability compared to previous datastore focusing on singular domains like Wikipedia. Regarding the flexibility in updating the datastore, it is simple to add / remove new text corpus by adding / removing embeddings with the backbone FAISS index without having to rebuild the whole index.
>
> **Q2. Does Consistency of the Retrieved Results Significantly Impact Performance**: We conduct an extra analysis on the top 10 retrieved passages for MMLU Pro. On average across all queries, the percentage of the overlapping passages is only 0.11. Among the overlapping passages, the spearman correlation is 0.079, and the kendall’s tau is 0.076. These results suggest that rankings in cosine similarity between the query and retrieved passages using $E_{Approx}$ and $E_{Exact}$ are largely different, which could account for the performance gap before and after the second stage Exact Search when k=10 as shown in the table below:
> | ANN ($E_{Approx}$) | Two-stage ($E_{Exact}$) |
> |---|---|
> | 50.1 | 53.1 |
>
> **Q3. Quantitative Analysis of Retrieval Quality**: As the evaluation queries are complex questions beyond finding a target piece of information, there is no set of retrieval results that are “ground-truth” to a given query, so we cannot calculate the precision / recall of our retrieval results. We motivate this as an open problem (since validation of ground-truth or “oracle” documents is an expensive process) to find oracle documents for modern benchmarks, which would enable another axis of measuring retrieval quality on a broader swath of benchmarks.
>
> In place of this, in Appendix E, we have included qualitative examples of retrieved documents in different stages and showed that retrieved documents from stronger retrieval strategies tend to be more helpful. For example, in Table 29, ANN yields a retrieval result that is only tangentially related to the question at hand, while after applying LM-reranking, we are given a top document that explicitly details a key step in solving the problem.
>
> **Q4. Whether Retrieval Mainly Aids Factual Grounding or Intermediate Reasoning Steps**: From the qualitative analysis we include in Appendix E, we show that retrieved passages can help both on factual grounding by providing relevant factual information, and on intermediate reasoning steps by providing guidance on how to solve similar question, depending on the specific question and retrieved passages. At scale, determining whether a document aids in factual grounding and/or intermediate reasoning unfortunately becomes quite expensive due to needed manual human evaluation or a strong LLM-as-judge.

---

> ### Author Response · Authors · 2025-11-20
> **Part 2/2 of the Response to Reviewer Uhfh**
>
> **Q5. Experiment with A Fine-tuned Retriever**: We have tried ReasonIR[1], a fine-tuned retriever trained for reasoning tasks in our preliminary experiments for exact search stage. We observed an average downstream performance of 55.3 with ReasonIR compared to an average of 56.5 with GritLM on our five benchmarks, so we decided to use GritLM in our pipeline.
>
> ---
>
> [1] ReasonIR: Training Retrievers for Reasoning Tasks (Shao et al., 2025)

---

> > ### Comment · Reviewer_Uhfh · 2025-11-23
> >
> > Thank you for the responses. I'm satisfied with the clarifications and will keep my positive rating.

---

### Official Review · Reviewer_GFMa · 2025-11-02

**Soundness:** 3
**Presentation:** 3
**Contribution:** 2
**Rating:** 6
**Confidence:** 3

**Summary:**

The authors claim that past works have shown little success in leveraging minimal RAG pipelines to improve performance on reasoning-intensive benchmarks because there are no practical, web-scale databases that capture the breadth of the pretraining distribution. At the same time, a RAG database must be compact to minimize decoding latency. The authors introduce CompactDS to meet these constraints. CompactDS is a curated mix of data that is initially drawn from a wide variety of sources and filtered to reduce redundancy and data contamination. In addition, the authors introduce a two stage retrieval pipeline: a course-grained lossy nearest neighbor search to narrow down the candidate set of passages followed by an exact inner product search conducted on-device to minimize retrieval latency.

**Strengths:**

-	Well written and motivates why current RAG pipelines are insufficient due to lack of scope and high retrieval latency
-	Demonstrate improvement on a variety of benchmarks using RAG
-	Considers problems in actual deployment (ie on device fine grained search after coarse-grained candidate search to minimize latency)
-	Upper-bounding the RAG performance with the oracle baseline makes sense and demonstrates how close this “autonomous” RAG system is to a curated ICL baseline

**Weaknesses:**

-	The paper introduces CompactDS as a means to improve reasoning performance using RAG but focuses mainly on multiple-choice question style evals that aren’t particularly reasoning intense. The MATH results are particularly interesting, but I would have liked to see the upper bound performance on this task (few shot prompting/ICL). My main concern is that the authors claim RAG helps with reasoning performance, but it seems hard to separate the boost from fact retrieval from the boost in reasoning performance, especially on MCQ style questions with non-reasoning models (it seems like in Table 4, retrieval seems to hurt GPQA performance across the board).
-	Is there a better way to gauge the accuracy of the two-stage retrieval pipeline beyond comparing downstream performance to the oracle? I am concerned because the oracle passages are chosen from the initial pool provided by the lossy retrieval.

**Questions:**

-	When would you use RAG over hand-curating examples per topic? You could curate examples for each of these baselines that might boost performance further. Is the claim that RAG allows an agent to operate autonomously? Could you illustrate what a system like this would look like? The total dataset size is still hundreds of GB of RAM, and the course grained search probably can’t be done on a small edge device.
-	Are you concerned that filtering heavily for contamination directly conflicts with the goal of RAG (retrieving relevant passages).
-	What would MASSIVEDS performance look like (efficiency and accuracy) when using the two stage retrieval pipeline?

---

> ### Author Response · Authors · 2025-11-21
> **Part 1/2 of the Response to Reviewer GFMa**
>
> We thank the reviewer for their helpful feedback, and for supporting our work!
>
> **W1. Are MC questions sufficiently reasoning-intensive?**: We want to point out that the multiple-choice questions in our evaluation can still require substantial reasoning to answer (We include the example from Table 30 as well as another complex example below). In Appendix E, we include detailed qualitative analysis that shows that simple “fact retrieval” is not sufficient to answer many of the questions and our retrieved documents can provide guidance on how to reason through similar questions.
>
> Here are two example of from MMLU Pro that requires complex reasoning:
> ```
> Question: Suppose a convex 3d-object has 15 vertices and 39 edges. How many faces does it have?
> ```
> ```
> The assessed valuation of the taxable property in the town of Smithville is $52,384,600. The taxes to be raised are $123,475 for a new local project, $931,442.75 for educational purposes, and $319,878 for health and welfare needs. Find the town's tax rate (a) to thousandths of a percent, (b) in mills per $1 of assessed value (c) in cents per $100 of assessed value, and (d) in mills per $1,000 of assessed value.
> ```
>
> **W1. Is No-retrieval Baseline for MATH Strong Enough?**: For MATH, we conduct an experiment with Llama 3.1 8B Instruct with 4-shot examples in context following Tulu3:
> | 0-shot | 4-shot |
> |---|---|
> | 46.9   | 46.7   |
>
> Here, we not only see that our 0-shot baseline is close to the potential upper-bound given by handcurated few-shot examples, but it actually is marginally higher.
>
> **W1. Reasons for Weaker Performance on GPQA**: For GPQA results in Table 4, we would like to point out that retrieval improves the performance on average for Llama 3.1 8B Instruct and Mistral 7B Instruct, and hurts the performance with Qwen3 8B and Llama 3.3 70B Instruct. We suspect it is because (1) the model loses capability to properly utilize the retrieved results after CoT training and (2) as GPQA performance rises, closing the gap requires more specialized, graduate-level knowledge that our web-based datastore may not fully cover.
>
> **W2. Gauging the Accuracy of the Retrieval Pipeline**: We first clarify that our oracle performance is only to serve as an analysis of an approximate upper-bound of the potential of CompactDS (e.g., as retrievers become more capable). Secondly, we believe there is no easy way to measure retrieval quality because these benchmarks have rarely been explored under the lens of RAG; no true oracle documents have been identified for the problems across these benchmarks. However, we agree the problem of finding true oracle documents for a benchmark (documents that we are sure are helpful for solving the given problems) is an important orthogonal, open challenge for the reason the reviewer mentioned: it unlocks another axis of measuring general retrieval quality for a breadth of different benchmarks. A first, naive step to tackling this challenge could simply be increasing the breadth of the first stage retrieval and conducting oracle-reranking over a larger number of documents (at a greater cost). For non-MC questions, this might involve using LLM-reranking where the reranked LLM is much stronger and the answer is provided to aid in ranking. An alternative approach could be to instead generate synthetic oracle documents beforehand for each problem and then inject them like canaries into the datastore. Ultimately, while important to study, we believe this question is orthogonal to our main goal, which is to demonstrate that retrieval is beneficial to downstream performance on these benchmarks.

---

> > ### Author Response · Authors · 2025-11-21
> > **Part 2/2 of the Response to Reviewer GFMa**
> >
> > **Q1. Why not use Handcrafted Examples Over RAG**: Firstly, as we have shown in the experiment for W1, using examples brings marginal performance gains over 0-shot, which is much less than the performance gain (+9.0%) by using CompactDS. Secondly, in addition to what the reviewer has mentioned about RAG enabling these systems to attain helpful knowledge autonomously, we emphasize that retrieval can help gather a greater breadth of helpful documents. For example, retrieval could provide documents with specific knowledge (e.g., a theorem) that must be used in conjunction with reasoning, as well as documents demonstrating how to exactly apply this specific knowledge, yielding examples that can be even more specific than hand-curated examples, especially at scale in autonomous systems like the reviewer mentioned.
> >
> > While we agree that $\mathcal{O}(100)$ GBs of RAM is still a non-trivial requirement, especially with the intention of hosting on edge devices, we still believe it is useful for easy deployment from a server setup or personal workstation (which is particularly beneficial for academic research). Because we’ve greatly reduced the requirement from tens of TB of RAM to $\mathcal{O}(100)$ GBs of RAM, we believe this will make pretraining-corpora-breadth search more accessible. Furthermore, edge devices don’t need to host the datastore on-device. For example, we are also releasing an API-queryable version of CompactDS; an edge device could then interface with CompactDS through API.
> >
> > **Q2. Conflict between Decontamination and RAG**: From a higher level, the goals of decontamination and retrieval do not conflict. Decontamination ideally serves to remove documents that contain the exact test questions and answers such that if they were retrieved, solving the problem would be trivial and demonstrate no generalizability. On the other hand, retrieval aims to get documents that contain helpful, relevant knowledge and ideas that can be used to help solve the problem. We decontaminate our datastore to, as much as possible, remove these documents that exactly contain the test questions and answers, while retaining helpful documents with useful knowledge/ideas. This follows a similar motivation for why we decontaminate training corpora before training. Empirically, however, we agree and observe that too strict of a decontamination can hurt RAG performance, likely due to removing too much helpful information even if those documents didn’t exactly contain test questions/answers.
> >
> > **Q3. MassiveDS under the Two-stage Pipeline**: Due to the scale of MassiveDS, we are unable to reproduce it and evaluate efficiency and performance directly. However, since MassiveDS uses a single-stage exact search, we’d expect the performance under an ANN first stage to drop slightly. However, this will enable MassiveDS to use a stronger model (e.g., GritLM-7B as we did) for exact search in the second stage, which could lead to potentially performance gain at the end. We still expect a lower overall performance than CompactDS due to the higher quality despite the smaller size.
> > On the other hand, the efficiency of MassiveDS would be greatly boosted. Due to its requirement of 12TB of RAM, MassiveDS in practice has to be split in parts and loaded into RAM sequentially, adding lots of overhead. With an ANN index, MassiveDS with ~3x more vectors than CompactDS has the potential to fit into 1TB of RAM with a set of more efficiency-prone parameters as we demonstrated in Appendix C.5, which would allow the entire index to be loaded into a machine at once. However, it would ultimately still be less efficient than CompactDS due to its larger size.

---

### Official Review · Reviewer_DAeh · 2025-11-12

**Soundness:** 2
**Presentation:** 2
**Contribution:** 2
**Rating:** 4
**Confidence:** 4

**Summary:**

This paper introduces CompactDS, a diverse datastore retrieval. The paper discusses the pipeline on generating this datastore and the method to efficiently serve the system. The paper claims that LLMs can perform better on reasoning-intensive tasks with RAG on this datastore.

**Strengths:**

1. The paper release a datastore for retrieval.
2. The paper includes a method to efficiently retrieve passages that reduce RAM needed.
3. The experiments are conducted on different model sizes.

**Weaknesses:**

## Improvements on reasoning-intensive benchmarks
I am not sure if the experiments fully support the claim, especially claiming that the proposed RAG system improves **reasoning extensive** tasks.
1. MMLU and MMLU Pro are more knowledge related. These two benchmarks ask more about model's knowledge. I am not familiar with AGI Eval but from Table 10 it seems like it is MC and no CoTs are allowed when the model answers. While the model still needs to reason internally even without CoT, I think "reasoning-intensive" means those benchmarks where the model need to produce very long CoT (?) Usually those are math and coding benchmarks like AIME, AMC, HMMT, LiveCodeBench for reasoning models.
2. The author includes MATH and GPQA where CoTs are allowed. While there are some improvement, how much improvement comes from the retrieval? For non-reasoning model like Llama-3 8B, perhaps a better baseline is to use a fixed, detailed prompt curated by a strong model like GPT-5 that teaches the model to plan, reflect, backtrack (and maybe some examples).
3. There should be more experiments on reasoning models. The paper includes Qwen3-8B in Table 10 but I am confused about the result (see Questions section below). Reasoning models achieve non-trivial scores in benchmarks like AIME so I wonder how does reasoning model perform on those challenging benchmarks.

## Other clarifications needed.
Please see questions below.

**Questions:**

1. In Table 4, Qwen3-8B has MATH accuracy 56.3. However, from Qwen3's technical report (https://arxiv.org/abs/2505.09388) the accuracy for MATH-500 is 97.4 (with reasoning; Table 17) or 87.4(without reasoning; Table 18). While MATH-500 is a subset of MATH, I wonder why there is such discrepancy?
2. In Table 10, what does "chat format" mean? Does it mean that the first three evals are selected based on model's perplexities on the choices and the last two are answered in QA?
3. Table 7 is confusing. Why the upper half table has MATH-500 "No Retrieval" accuracy 83.2 but lower half table with MATH-500 "No Retieval" accuracy 91.0?

---

> ### Author Response · Authors · 2025-11-21
> **Part 1/2 of the Response to Reviewer DAeh**
>
> We thank the reviewer for their helpful feedback, and for supporting our work!
>
> **W1. What Defines Reasoning-Intensive Benchmarks?**: We would like to first point out that there are many questions in MMLU and MMLU Pro requiring substantial reasoning beyond simple knowledge testing (we include the example from Table 30 as well as another complex example below). The reviewer is correct that MMLU, MMLU Pro, and AGI Eval are in multiple-choice format, and GPQA and MATH allow free-form generation. However, it is worth noting that long-CoT is a test-time scaling method that has been shown to be effective on some reasoning benchmarks, especially those math and coding benchmarks that the review has mentioned, rather than a restriction that defines a “reasoning-intensive” benchmark. For example, BRIGHT[1] is for evaluating retrieval only and fully decouples from LM generation, but it is widely-considered as a reasoning-intensive benchmark. It is still important to evaluate the model’s reasoning ability when not using test-time scaling or when the model is not trained to produce long-CoT.
>
> Here are two example of from MMLU Pro that requires complex reasoning:
> ```
> Suppose a convex 3d-object has 15 vertices and 39 edges. How many faces does it have?
> ```
> ```
> The assessed valuation of the taxable property in the town of Smithville is $52,384,600. The taxes to be raised are $123,475 for a new local project, $931,442.75 for educational purposes, and $319,878 for health and welfare needs. Find the town's tax rate (a) to thousandths of a percent, (b) in mills per $1 of assessed value (c) in cents per $100 of assessed value, and (d) in mills per $1,000 of assessed value.
> ```
>
> **W2. Room for a Potentially Better Prompt**: First, we note that we did our best to use the state-of-the-art prompts from prior work., e.g., we adopted the prompts [2] from Tulu 3 for GPQA and Llama 3.1 for MATH. Second, we note that both the llm-only and the retrieval evaluations have a prompt that elicits CoT in response, so the improvements observed are already directly tied to the information gain from prepending retrieved documents. One can possibly design prompts to elicit reasoning behaviors like planning / reflecting / backtracking; however, we believe they are orthogonal to the improvements from using CompactDS.
>
> **W3. Experiments with More Reasoning Models**: We highlight that we do include results using another notable reasoning model QwQ 32B, with results shown in Table 7. We evaluate the performance on the same benchmarks included in QwQ’s paper[3] (GPQA Diamond and MATH 500) with CompactDS and show that it consistently brings significant gains.
>
> We also cautiously approach testing with more modern reasoning models such as those under the Qwen family because many recent works have observed considerable test set contamination due to memorization of problems from benchmarks like MATH and AIME [4].
>
> ---
>
> [1] BRIGHT: A Realistic and Challenging Benchmark for Reasoning-Intensive Retrieval (Su et al., 2024)
>
> [2] OLMES: A Standard for Language Model Evaluations (Gu et al., 2024)
>
> [3] QwQ-32B: Embracing the Power of Reinforcement Learning (Qwen Team, 2025)
>
> [4] Reasoning or Memorization? Unreliable Results of Reinforcement Learning Due to Data Contamination (Wu et al., 2025)

---

> ### Author Response · Authors · 2025-11-21
> **Part 2/2 of the Response to Reviewer DAeh**
>
> **Q1. Discrepancy in Qwen3-8B Results**: We believe the difference in performance is likely due to differences in generation parameters and prompting (especially since we are using Qwen3-8B in reasoning mode), especially the much lower max output length we specified for our MATH/GPQA evaluations, *2048 tokens*. To the best of our knowledge, the values reported in the technical report used a *max output length of 32k*. Ultimately, we believe that studying evaluation under smaller output token budgets is also valuable for limited compute environments/token-efficient responses, as well as being more compute-efficient for evaluation. We note that our reported results with QwQ-32B use 32k output length, indicating gains with retrieval still hold at longer output token budgets.
>
> **Q2. Clarification on Definition of Chat Format**: Chat formats (also known as chat templating) refers to the specific formatting of the inputs when the model is post-trained. Models are post-trained on different chat formats, yielding another variable that sometimes affects performance, so it is common that post-trained models (e.g. Llama3) chooses not to use chat format for some evaluation benchmarks.
>
> *An example of questions not put in chat format*:
> ```
> In how many ways can 8 people be seated around a square table with 2 people on a side?
> (model’s output…)
> ```
> *An example of questions put in chat format*:
> ```
> <|user|>
> In how many ways can 8 people be seated around a square table with 2 people on a side?
> <|assistant|>
> (model’s output…)
> ```
>
> **Q3. Confusion around values reported in Table 7**: The upper half of the table shows results exactly as reported from the Search-o1 paper [5], whereas the lower half is results from our evaluation. We will make sure this distinction is clearer in the next revision.
>
> ---
>
> [5] Search-o1: Agentic Search-Enhanced Large Reasoning Models (Li et al., 2025)

---

### Author Response · Authors · 2025-11-21

We thank all reviewers for their time and effort in reviewing our submission. During the author response period, we provide analysis, explanations, and additional results to address the comments from the reviewers. We provide a summary below of recognized contributions and common concerns:

**Recognized Contributions**

- (Reviewers GFMa and Uhfh) The motivation of our work to build a high-quality, web-scale, and diverse datastore for enabling RAG to perform effectively on reasoning-intensive benchmarks is well-grounded and clear.
- (Reviewers DAeh, GFMa, 41Bx) CompactDS considers problems in actual deployment by implementing a two-stage pipeline to efficiently retrieve passages that significantly reduces the compute resources necessary compared to earlier works in pretraining-breadth datastores.
- (Reviewers DAeh, GFMa, Uhfh, 41Bx) A broad set of experiments across multiple model families and evaluation settings, including grounded comparisons with popular approaches such as search engines/agentic systems, demonstrate the general applicability of CompactDS.
- (Reviewers DAeh, Uhfh, 41Bx) The release of CompactDS is useful to the community as an open, reproducible, and accessible resource.

**Clarifications for Common Concerns**
- **Are our evaluation benchmarks sufficiently reasoning-intensive?** In response to reviewers DAeh and GFMa, we explain that the multiple-choice style of questions from some of our benchmarks do not undermine the fact that these questions require intensive reasoning to solve, showing two concrete example questions from the benchmarks. We also emphasize the point that reasoning-intensive benchmarks are not restricted to the ones that long CoT generations are most effective on, and evaluating models’ capabilities without test-time scaling methods is still important.
- **Is CoT or better prompting the only contributing factors of good reasoning performance?** Reviewers DAeh and 41Bx bring up the question about whether reasoning-oriented prompting or training can yield better baseline reasoning performance that makes the impact of retrieval negligible. We clarify that, while these techniques are beneficial to reasoning performance, they are orthogonal to the gains from using retrieval on our task suite. For example, in all our evaluations where CoT is applicable for both LLM-only and with retrieval, we use a prompt specifying the use of CoT and our reasoning models have thinking enabled. Thus, the performance gains are solely from the addition of the retrieved documents.
- **More justification behind datastore curation choices** Reviewer 41Bx raises an important point about clarifying our datastore curation decisions to build greater intuition over what makes a datastore good. We provide a summary below:
  - Our pipeline largely follows best practices in modern LLM data curation, and draws heavily from recent high-quality open-source training efforts.
  - Web data: We included the web data as it has been the de-facto source for broad coverage and diversity. We use both C4 and DCLM-Baseline, because they’re known to be the highest-quality Common Crawl based datasets available. However, both rely on relatively simple heuristic and fastText-based filtering. For this reason, we apply another layer of stronger quality filtering based on the FineWeb-Edu classifier.
  - Wikipedia and Books: We include them because they have long been recognized as high-quality corpora for pretraining. For instance, BERT is trained on Wikipedia and Books.
  - Expert Data: We include expert domains such as math, academic papers, and code, as strong reasoning capabilities often require exposure to these domains. For each domain, we use the most up-to-date, carefully curated public datasets.
  - Q&A forums: We included Q&A forums as these sources are widely regarded as high-quality for training and often contain well-structured problem-solution pairs.

  We select these datasets with the goal to compose a high-quality, diverse, and large-scale datastore. We include more experiments to show that the diversity of CompactDS is critical, and the high quality of it contributes to the efficiency in Sec 4.2 and Appendix C.1 in our paper.

---

### Meta-Review · Area_Chair_5kU6 · 2026-01-07

**Summary:**

This paper builds CompactDS, a curated web-scale datastore with a simple two-stage retrieval setup, and shows RAG can improve results on benchmarks like MMLU, MMLU Pro, GPQA, and MATH. Reviewers agree the work is clearly written, practical, and the release could be useful. However, the main claim about “improving reasoning”could be more convincing if more benchmarks beyond multiple-choice/knowledge heavy QA are used. Some design choices in data filtering and the retrieval pipeline still feel under-justified. Overall, I recommend boardline acceptance.

**Reviewer Concerns:**

- Rebuttal addressed: (1) deployment practicality, (2) evaluation protocol clarity.

- Still outstanding are: (1) the “reasoning improvement” claim remains hard to disentangle from knowledge/fact lookup on largely MCQ-heavy suites, (2) limited quantitative evidence of retrieval quality beyond downstream scores/oracle-from-limited-pool.

**Reviewer Scores:**

- DAeh: likely a small raise after clarifying eval settings/tables, but still skeptical about the “reasoning” framing and baselines.
- GFMa: likely unchanged or slight raise ; rebuttal addressed most concerns, with GPQA still a caveat.
- Uhfh: unchanged; they explicitly said the clarifications resolved their concerns.
- 41Bx: likely a modest raise, but probably still below threshold due to mixed GPQA and limited novelty/evidence separation.

---

### Decision · Program_Chairs · 2026-01-26

Accept (Poster)